# DeStyle2Style: Scalable Destylization-Driven Data Generation for Artistic Style Transfer

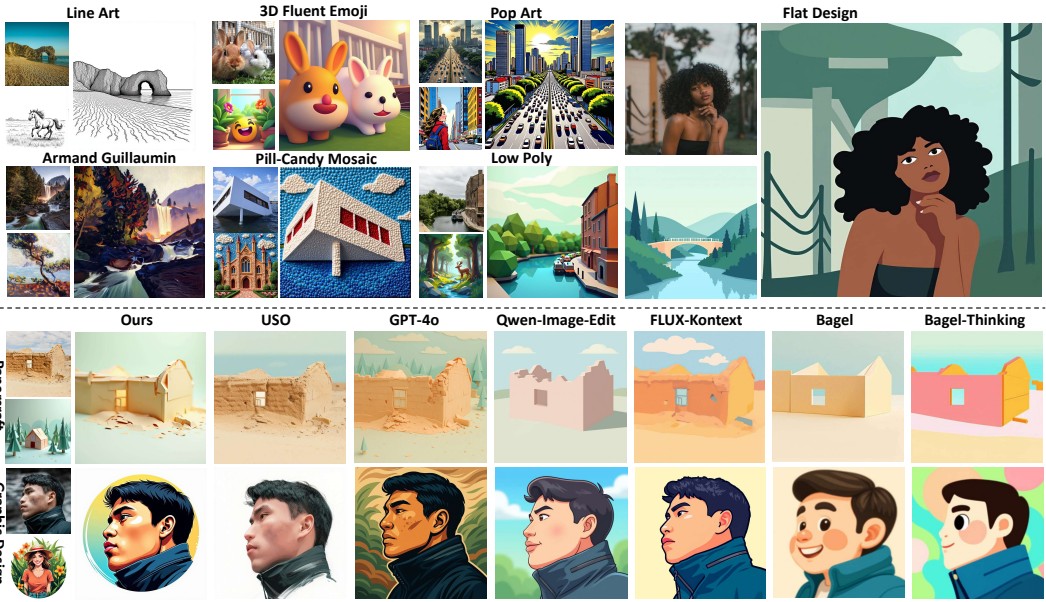

Figure 1: The top part shows our style transfer results across diverse artistic styles at 1K resolution, while the bottom part presents comparisons between our method and existing image editing models.

## ABSTRACT

DeStyle2Style introduces a novel approach to artistic style transfer by reframing it as a data problem. Our key insight is destylization, reversing style transfer by removing stylistic elements from artworks to recover natural, style-reduced counterparts. This yields DeStyle-100K, a large-scale dataset that provides authentic supervision signals by aligning real artistic styles with their underlying content. To build DeStyle-100K, we develop DestyleNet, a text-guided destylization model that reconstructs style-reduced natural images, and DestyleCoT-Filter, a multi-stage evaluation model that employs Chain-of-Thought reasoning to automatically discard low-quality pairs while ensuring content fidelity and style accuracy. Furthermore, we introduce BCS-Bench, a benchmark with balanced stylistic diversity and content generality for systematic evaluation of style transfer methods. Our results demonstrate that scalable data generation via destylization offers a reliable supervision paradigm, effectively addressing the fundamental challenge of lacking "ground-truth" data in artistic style transfer.

## 1 INTRODUCTION

Style transfer (Gatys et al., 2016), which aims to modify an image's stylistic appearance while maintaining its underlying content, has attracted widespread interest for its applications in creative fields such as digital art, advertising, and fashion. Over the years, style transfer techniques have progressed rapidly, moving from early optimization-driven methods (Gatys et al., 2016; 2017; Kolkin et al., 2019) to more recent diffusion model-based solutions (Wang et al., 2024a;b; Xing et al., 2024; Junyao et al., 2024; Sohn et al., 2023).

While style transfer has made significant progress in recent years, it remains fundamentally ill-posed, as there exists no definitive "ground-truth" stylization for a given content–style pair. Most prior works attempt to address this challenge from a model-centric perspective, ranging from early efforts using VGG-based feature statistics (Gatys et al., 2015; 2016; Zhang et al., 2019; Kolkin et al., 2019; Gatys et al., 2017), to recent advances based on diffusion model fine-tuning (Sohn et al., 2023; Frenkel et al., 2024; Ouyang et al., 2025; Shah et al., 2023; Wang et al., 2025a) and inversion-based techniques (Chung et al., 2024a; Zhang et al., 2023a; Voynov et al., 2023) to circumvent the lack of definitive "ground-truth" supervision. However, such approaches still suffer from inaccurate style representation and uncontrollable optimization behaviors, owing to the absence of explicit supervision. This highlights the need for a data-centric solution that provides reliable stylization supervision. OmniStyle (Wang et al., 2025b) takes the first step toward data-centric supervision by synthesizing large amounts of stylized outputs using existing style transfer models and filtering them with multimodal LLMs (MLLMs), thereby constructing the first large-scale paired dataset OmniStyle-1M for style transfer. However, the synthesized results inevitably provide pseudo-supervision, as the supervision quality is fundamentally limited by existing style transfer models, resulting in unreliable and unauthentic approximations that fail to achieve consistent and faithful style transfer.

In this paper, **DeStyle2Style** also adopts a data-centric perspective, but follows a fundamentally different and more essential path, *destylization*. The destylization paradigm, instead of synthesizing stylized images from scratch, reverses the process by automatically reducing style information and extracting structure-aligned natural content images from real artistic artworks. This paradigm fundamentally addresses the core limitations of OmniStyle (Wang et al., 2025b) by enabling original artistic images to serve as the sole authentic supervision signals. Here, "authentic supervision signals" refer to using unaltered style images as direct learning targets, rather than relying on synthetic data generated through style transfer models that are modified from existing images. By doing so, the supervision signals are derived exclusively from high-quality original style images, while the de-stylized images serve solely as content inputs, ensuring that the supervision quality remains uncompromised. On the contrary, their minor imperfections naturally introduce beneficial variations, effectively serving as data augmentation to improve model robustness. Specifically, the core of DeStyle2Style is a text-guided destylization model, **DestyleNet**, which leverages accompanying textual descriptions to guide the reconstruction of natural, style-reduced counterparts from artistic inputs. Leveraging this approach, we are able to extract style-reduced, structure-aligned natural content from a wide range of real and origin artistic images, enabling the construction of a reliable and diverse dataset. Consequently, we construct **DeStyle-100K**, a high-quality dataset comprises 100K high-quality image triplets in the form of $\langle$ **de-stylized image**, **reference image**, **style image** $\rangle$[1]. As shown in Figure 2, the dataset encompasses a diverse range of visual styles, including traditional artworks from 669 renowned artists (e.g., Van Gogh and Monet) across 117 art movements (e.g., Impressionism, Baroque), as well as 65 mainstream digital styles such as origami art, 3D, flat design, line-art, ink painting, and others. To ensure data quality, we further introduce **DestyleCoT-Filter**, a Chain-of-Thought-based filtering mechanism that evaluates the plausibility of the destylized image as a natural, style-reduced counterpart along two dimensions: content preservation and style discrepancy. Unlike prior approaches that directly apply MLLMs to assess stylized outputs, which often involve complex and subjective artistic attributes, DestyleCoT-Filter operates on destylized images that better align with the training distribution of MLLMs. This makes the evaluation more robust and reliable. In addition, DestyleCoT-Filter employs a multi-stage, fine-grained assessment framework that facilitates interpretable and controllable quality filtering. Finally, to enable comprehensive evaluation, we introduce **BCS-Bench**, which consists of 56 style images across 35 representative styles and 55 content images spanning six major content categories: human, animal, plant, scene, architecture, and object. These form a total of 3,080 content-style pairs for systematic evaluation.

Our contributions include **1) DeStyle2Style** reframe artistic style transfer as a data generation problem, it enables the use of unaltered style images as direct learning targets through de-stylization, providing high-quality supervision signals for the style transfer. DeStyle2Style demonstrates that scalable and high-quality supervision via destylization is key to achieving reliable and faithful style transfer. **2)** We introduce **DeStyle-100K**, a large-scale dataset of 100K high-quality triplets constructed through destylization. Unlike prior pseudo-target datasets, DeStyle-100K provides *authentic supervision*, where unaltered style images directly serve as training signals through a reverse formulation. **3)** We develop **DestyleNet**, a text-guided destylization model capable of reducing di-

---

[1] green:input; blue:"ground-truth"

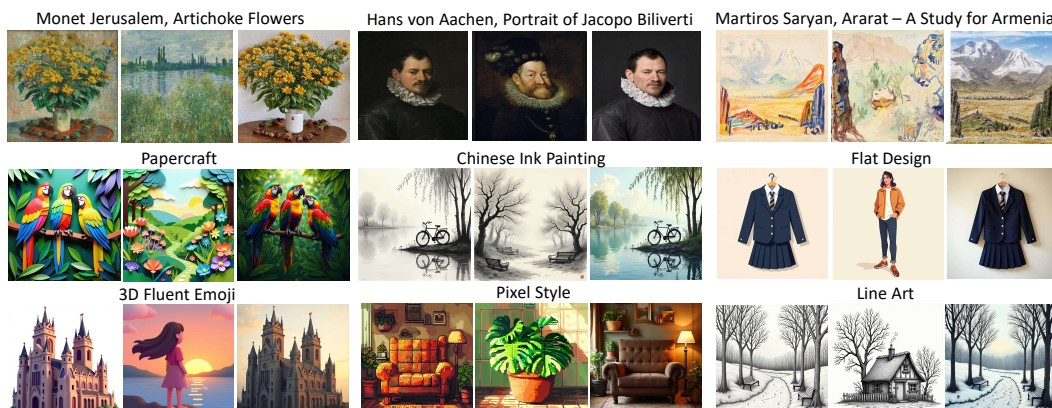

Figure 2: **Representative Samples of DeStyle-100K.** DeStyle-100K consists of 100K high-quality triplets in the form of ⟨style image, reference image, de-stylized image⟩, covering classical artistic styles from 669 artists across 117 art movements, and supporting 65 mainstream digital styles. More samples can be found in the appendix.

verse artistic styles while faithfully preserving structure-aligned content. To ensure data integrity, we design **DestyleCoT-Filter**, a fine-grained CoT-based evaluation framework that enforces both content preservation and style discrepancy. **4)** We propose **BCS-Bench**, a benchmark with balanced stylistic diversity and content generality for systematic evaluation of style transfer methods. It consists of 56 style images spanning 35 representative artistic styles and 55 content images covering 6 major semantic categories (human, animal, plant, scene, architecture, object), forming 3,080 diverse content-style pairs for quantitative and qualitative analysis.

## 2 RELATED WORK

**Style Transfer.** Style transfer has advanced rapidly, evolving from handcrafted features and filter-based stylization (Zhang et al., 2013; Wang et al., 2004), to optimization-based approaches (Gatys et al., 2016; 2017; Kolkin et al., 2019), and then to feed-forward models enabling arbitrary transfer (Huang & Belongie, 2017; Li et al., 2017; Liao et al., 2017; Zhang et al., 2022a; Deng et al., 2020). Recently, diffusion-based methods (Wang et al., 2024a; Chung et al., 2024b; Xu et al., 2024; Xing et al., 2024) have further pushed performance, through both tuning-based (Zhang et al., 2023b;a; Wang et al., 2023) and tuning-free (Wang et al., 2024b; Junyao et al., 2024; Qi et al., 2024) paradigms. Despite these advances, a fundamental limitation remains: the lack of definitive "ground-truth" for stylization, which hinders supervised training. Existing methods rely on hand-crafted metrics, unstable inversion, or pseudo-supervised fine-tuning (Wang et al., 2025b), resulting in noisy learning signals and weak style representations. To address this, we propose a novel destylization paradigm that reverses the stylization process to extract style-reduced and structure-aligned content from style images. This enables the construction of grounded content–style supervision pairs. Based on this, we introduce DeStyle-100K, a high-quality dataset created via destylization, providing authentic supervision for training style transfer models.

**Datasets for Style Transfer.** Early style transfer datasets, such as WikiArt (Tan et al., 2019) and Style30K (Li et al., 2024), provide artistic exemplars but lack aligned triplets, making them unsuitable for supervised training. Recent efforts (Xing et al., 2024; Wang et al., 2025b) attempt to construct synthetic triplet datasets, but their quality is limited by the performance and biases of the underlying style transfer models. Although MLLMs are used for filtering, their reliability on stylized images remains questionable due to limited domain understanding. As a result, the supervision may be noisy, with style drift, artifacts, and poor generalization. In contrast, we propose a destylization-based construction pipeline that reverses the stylization process to recover natural content, allowing MLLMs to perform reliable evaluation. This enables the creation of triplets with accurate content alignment and authentic style supervision.

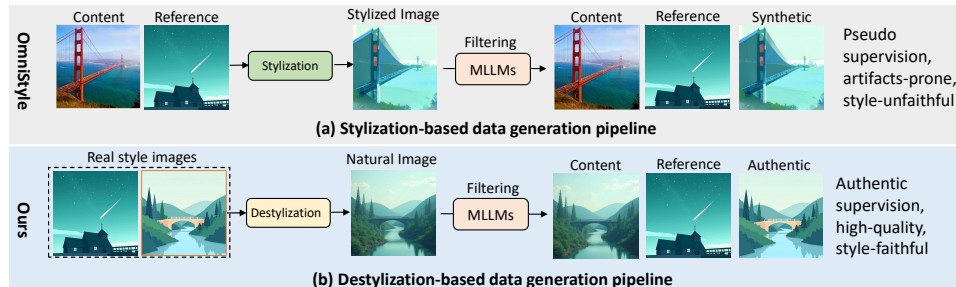

Figure 3: (a) Stylization-based data generation pipeline (OmniStyle). (b) Destylization-based data generation pipeline (ours). Our method enables authentic supervision with high-quality and style-faithful data, in contrast to stylization-based pipelines that rely on pseudo-supervision, often artifacts-prone.

## 3  METHOD

In this section, we first compare the advantages and limitations of two data construction pipelines: our proposed destylization-based pipeline and the stylization-based pipeline of OmniStyle (Section 3.1). We then introduce the design of DestyleNet (Section 3.2), followed by a detailed description of how we construct a DeStyle-100K dataset (Section 3.3). Next, we introduce DestyleCoT-Filter (Section 3.4), a fine-grained evaluation mechanism for data quality control. Finally, we describe the overall architecture of DeStyle2Style and detail its training procedure (Section 3.5).

### 3.1  DESTYLIZATION VS. STYLIZATION

Stylization and destylization are inverse processes: while stylization aims to transfer artistic style onto a natural image, destylization seeks to reduce stylistic elements from an artwork to recover its underlying natural content. OmniStyle adopt stylization-based pipelines (see Figure 3.a), which generate synthetic stylized results by applying style images to content images using pre-trained style transfer models. However, due to the limited capabilities of current style transfer models, such pipelines often suffer from visual artifacts, content leakage, and style inconsistency, resulting in pseudo-supervision that compromises the quality and fidelity of the constructed datasets.

In contrast, we propose a novel destylization-based pipeline (see Figure 3.b) that reverses this process: starting from real artworks, we reduce style using a dedicated destylization model to recover the underlying natural appearance. This enables the construction of training triplets in which the style transfer supervision is derived directly from style images, offering higher fidelity, authentic style supervision, better alignment with the original artistic distribution, and more faithful learning signals for style transfer. Authentic supervision, in our context, refers to supervision signals derived from unmodified style images rather than pseudo-stylized results synthesized by applying style transfer models to content images. Unmodified style images include real artworks and high-quality images synthesized from text prompts via FLUX-T2I (Black Forest Labs, 2024). We next provide a detailed introduction to our destylization approach.

### 3.2  DESTYLENET

DestyleNet is a text-guided destylization model that reduces stylistic attributes from a style image and generates a structure-aligned content image. In the following sections, we present the construction of the destylization dataset and the architecture of DestyleNet.

**Destylization Dataset.** To train the DestyleNet, we construct a dedicated dataset, as shown in Fig. 4(a). We first select 200 high-resolution content images for each of six semantic categories including humans, objects, animals, plants, scenes, and architectures from HQ-50K (Yang et al., 2023) and FFHQ (Karras et al., 2019). For style references, we collect 200 classical paintings from the National Gallery of Art (National Gallery of Art, 2025) and 200 style images from Style30K (Li et al., 2024). Each content image is stylized using four state-of-the-art methods: STROTSS (Kolkin et al., 2019), StyleID (Chung et al., 2024b), CSGO (Xing et al., 2024), and Attention Distillation (Zhou et al., 2025), guided by style images. Content captions are generated using InternVL2.5-7B (Chen

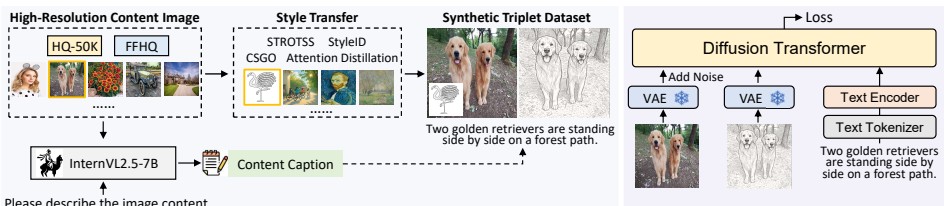

Figure 4: (a) Destylization Dataset Construction and (b) The architecture of DestyleNet model.

et al., 2024). This results in 60K stylized, content, and caption triplets for training the destylization model.

**DestyleNet Architecture.** Building upon the constructed triplet dataset, we design DestyleNet based on the FLUX-Dev (Black Forest Labs, 2024) model, as illustrated in Figure 4(b). The core idea of DestyleNet is to reduce stylistic information from the input style image under the guidance of a content text prompt. Specifically, we first employ a Variational Autoencoder (VAE) to extract continuous visual features from both the content image and its corresponding stylized image, while a text encoder is used to extract semantic features from the content caption. To obtain a style-reduced output, Gaussian noise is added to the visual features of the content image, which serves as the learning target. The stylized image features and text features are then spatially concatenated with the noisy content features to form a complete token sequence, which is subsequently fed into the FLUX DiT for image generation. During inference, DestyleNet takes as input a style image and its corresponding content caption, and produces a style-reduced natural image. As shown in Figure 2, DestyleNet demonstrates robust applicability across a wide spectrum of style domains. In addition to classical paintings, our model effectively reduces stylistic elements from diverse and complex art styles, including papercraft, 3D, pixel art , chinese ink painting, flat design, and line art, and more. This generalization ability provides essential model support for the construction of the DeStyle-100K.

### 3.3 DESTYLE-100K DATASET

Based on DestyleNet, we perform a two-stage destylization pipeline to construct the DeStyle-100K: (1) collecting a diverse set of style images and (2) conducting text-guided destylization.

**Style Images Collection.** To construct the DeStyle-100K dataset, we build a large-scale style image pool that incorporates both real and synthetic artworks with diverse stylistic attributes. For real images, we collect classical artworks from public datasets such as WikiArt (Tan et al., 2019) and the National Gallery of Art (National Gallery of Art, 2025), followed by a multi-stage filtering process to remove low-resolution, non-artistic, and duplicate images. We further apply InternVL2.5-7B (Chen et al., 2023) to retain images with concrete and interpretable scenes, categorize them into six content classes (Human, Animal, Plant, Object, Scene, Architecture), and discard stylistically ambiguous cases. This yields 10K high-quality real artworks spanning 669 artists (e.g., Van Gogh, Monet) and 117 movements (e.g., Impressionism, Baroque), all resized to 1024 × 1024. To compensate for the limited diversity and availability of real artworks, we synthesize additional stylized images using FLUX-Dev (Black Forest Labs, 2024). Specifically, we define a 65-category style taxonomy (e.g., Pixel Style, Cyberpunk, Line Art) and a hierarchical content tree with six top-level classes, each further divided into 10 subtypes (e.g., "Fantasy character", "Traditional Asian architecture"). For each style, we randomly pair it with 300 content subtypes to form diverse style–content combinations. We then employ GPT-4o to generate detailed joint prompts for each pair, and render 1024 × 1024 style images using FLUX-Dev with randomly sampled seeds, resulting in a total of 150K synthetic images.

**Text-Guided Destylization.** We use GPT-4o to generate content-focused descriptions of style images, explicitly instructed to ignore stylistic attributes and focus solely on plausible real-world semantics, such as object identity, scene type, pose, and spatial layout. These descriptions are then used as text prompts to guide the destylization process with DestyleNet, yielding a large number of style–destylized image pairs.

## 3.4 DESTYLECOT-FILTER

To ensure high-quality data, we introduce DestyleCoT-Filter, a Chain-of-Thought-based filtering mechanism that evaluates the quality of style–destylized image pairs. Unlike previous MLLM-based filtering methods (Wang et al., 2025b), which focus on assessing stylized results, DestyleCoT-Filter evaluates destylized images (i.e., natural-looking counterparts), making the assessment more robust. This avoids the need for complex domain knowledge of art history or stylistic conventions. The DestyleCoT-Filter pipeline consists of two complementary evaluation components: *content preservation* and *style discrepancy*, which together ensure that the destylized image retains the original content while effectively reducing the artistic style.

**Content Preservation.** Directly prompting GPT-4o to assess content consistency often fails to capture fine-grained mismatches. To address this, we adopt a Chain-of-Thought (CoT) strategy that guides GPT-4o to: (1) identify key semantic regions in the style image (e.g., faces, hands, text, scene elements); (2) verify their structural and visual consistency in the destylized image; and (3) assign a quality score from 0 to 5 based on the most significant failure, penalizing even minor omissions or distortions. Explanations are provided for each rating to enhance interpretability.

**Style Discrepancy.** To directly assess how much stylistic information is reduced, we adopt a fine-grained evaluation strategy that decomposes the style image into distinct attributes, such as color palette, texture, lighting, and rendering effects. GPT-4o is then guided to compare these attributes with the destylized result. We assign a 0–5 score reflecting stylistic reduction, accompanied by a brief rationale.

We evaluate all candidates for content preservation and style discrepancy, retaining only samples with both scores $\geq 4$. This yields 100K high-quality style–destylized pairs. For each style image, we compute the CSD (Somepalli et al., 2024) score over images in the same category and select the one with the highest stylistic similarity as the reference to form triplets.

## 3.5 DESTYLE2STYLE MODEL

Building upon the DeStyle-100K dataset, we propose DeStyle2Style, a simple yet effective style transfer framework based on FLUX-Dev (Black Forest Labs, 2024). Specifically, given a triplet of images in the form of style-reference-destylized, DeStyle2Style treats the style image as the denoising target. The reference image and the destylized image serve as conditional inputs to the DiT module, while the text input is left empty. All images are encoded into continuous visual features using a pretrained VAE. Gaussian noise is added to the features of the style image to construct a denoising training objective. To effectively model the transformation from the destylized to the style image, we introduce sequential positional encoding to the input tokens. This sequential encoding better captures the ordering and interaction within the triplet to avoid content confusion. Specifically, tokens extracted from the style, reference, and destylized images are assigned continuous and non-overlapping position indices, allowing the model to explicitly distinguish the role and order of each image in the style transfer pipeline. For efficient training, we adopt LoRA-based fine-tuning instead of full-model updating. This not only reduces memory overhead but also helps preserve the pretrained knowledge, leading to improved stylization performance.

Table 1: Comparison of existing style transfer benchmarks and our proposed BCS-Bench. "N/A" denotes missing information.

| Benchmark | Content Images | Content Categories | Style Images | Style Categories | Content-Style Pairs | Resolution |
|---|---|---|---|---|---|---|
| CAST (Zhang et al., 2022b) | N/A | N/A | N/A | N/A | 50 | N/A |
| AesPANet (Hong et al., 2023) | N/A | N/A | N/A | N/A | 65 | 256×256 |
| InST (Zhang et al., 2023b) | N/A | N/A | N/A | N/A | 26 | N/A |
| StyleID (Chung et al., 2024b) | 20 | 4 | 40 | Only Oil paintings | 800 | 512×512 |
| StyleShot (Junyao et al., 2024) | 20 | 6 | 490 | 73 | 9,800 | 879×876 |
| OmniStyle (Wang et al., 2025b) | 20 | 4 | 100 | 32 | 2,000 | 1024×1024 |
| BCS-Bench (Ours) | 55 | 6 | 56 | 35 | 3,080 | 1024×1024 |

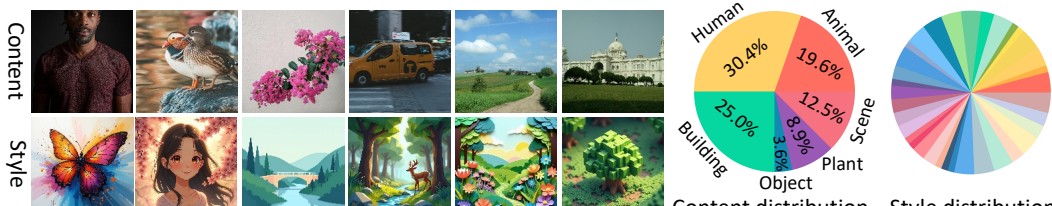

Figure 5: Content and style examples (left) and distributions (right) of BCS-Bench. For clarity, 35 style categories are omitted.

## 4 BENCHMARK AND EVALUATION

### 4.1 BCS-BENCH

As summarized in Table 1, previous benchmarks mainly emphasize stylistic diversity while providing limited and sometimes biased content coverage. For example, StyleShot contains 490 style images from 73 categories, but its content set covers only 20 content images and even includes stylized or non-natural images, which compromises its role as a clean content reference. Similarly, StyleID focuses primarily on oil paintings, while datasets such as AesPANet and InST are relatively small in scale and resolution. To address these limitations, we introduce BCS-Bench (see Figure 5), a curated benchmark that balances stylistic diversity and content generality. Specifically, BCS-Bench includes 56 style images spanning 35 representative artistic styles, ranging from common 2D styles (e.g., pixel art, flat design, sketch) to complex 3D-rendered styles (e.g., origami, voxel art), along with 55 content images covering six major semantic categories: human, animal, plant, object, scene, and architecture. Combining all possible content–style pairs yields 3,080 unique combinations, enabling both quantitative and qualitative evaluation across diverse scenarios. All images are provided at a high resolution of 1024×1024, ensuring more realistic and comprehensive assessment of style transfer models compared to prior benchmarks.

### 4.2 EXPERIMENTAL SETTINGS

**Baselines**. We compare our method against two groups of baselines:

**(1) Style Transfer Models.** We include 7 representative and state-of-the-art style transfer methods: OmniStyle (Wang et al., 2025b), Attention Distillation (AD) (Zhou et al., 2025), StyleID (Chung et al., 2024b), StyleShot (Junyao et al., 2024), CSGO (Xing et al., 2024), AesPA-Net (Hong et al., 2023), and STROTSS (Kolkin et al., 2019).

**(2) Closed-/Open-Source Image Editing Models.** We further compare our method with recent image editing models, including both closed-source and open-source approaches. Specifically, we evaluate GPT-4o (closed-source) and representative open-source models such as FLUX-Kontext (Batifol et al., 2025), Qwen-Image-Edit (Wu et al., 2025a), Bagel (Deng et al., 2025), Bagel-Thinking (Deng et al., 2025) and USO (Wu et al., 2025b). For GPT-4o and USO, we input both content and style images with an instruction to transfer style, while for the remaining single-image reference models we convert the style image into a textual prompt as the editing instruction.

**Evaluation Metrics.** We evaluate model performance in terms of content preservation and style similarity. For content preservation, we use a DINO-based structural similarity score and a CLIP-based image-text alignment score. For style similarity, we adopt the CSD score (Somepalli et al., 2024) and traditional style loss. In addition, we introduce three new metrics based on Qwen-VL-Max (Bai et al., 2023): (1) Qwen-Content-Score, measuring content similarity between the stylized and content images; (2) Qwen-Style-Score, measuring style similarity to the reference; and (3) Qwen-Aesthetic-Score, assessing the overall visual and aesthetic quality. Each score is obtained by prompting Qwen-VL-Max with the relevant image pair, returning a value from 0 to 10, with higher values indicating better results.

**Implementation Details.** Both DestyleNet and DeStyle2Style are built upon FLUX-Dev (Black Forest Labs, 2024). Given the difficulty of the destylization task, we fine-tune the entire DiT module when training the DestyleNet model. In contrast, DeStyle2Style only fine-tunes the LoRA modules. Both models are trained on 8×A800 GPUs with a learning rate of 1e-4. The batch size is set to 8 for

DestyleNet and 48 for DeStyle2Style. To enhance robustness in both destylization and stylization learning, we apply horizontal and vertical flipping as data augmentation during training.

## 4.3 QUANTITATIVE EVALUATION

Our quantitative evaluation consists of two parts: (1) comparison with existing style transfer methods, and (2) comparison with both closed and open-source image editing models.

**(1) Comparison with Style Transfer Methods.** As shown in Table 2, our method achieves the best performance on three style-related metrics: Style Loss, CSD Score, and Qwen Style Score. It also ranks second in Qwen Content Score and is among the top three in both DINO and CLIP Scores, demonstrating a strong balance between style fidelity and content preservation. While OmniStyle and StyleID yield slightly higher content scores, they often apply only minor color changes, leading to reduced style expressiveness. Notably, our method achieves the highest Qwen Aesthetic Score (8.7326), significantly surpassing all baselines and confirming its ability to generate visually appealing, high-quality stylizations.

**(2) Comparison with Closed and Open-Source Editing Models.** As shown in Table 3, GPT-4o achieves the best overall performance, ranking first across three metrics. DeStyle2Style consistently ranks second on multiple metrics, but still lags behind GPT-4o. USO exhibits low stylization strength (CSD Score 0.4441), which inflates its content score (DINO Score 0.8740) due to insufficient stylization. For open-source models such as Qwen-Image-Edit, FLUX-Kontext, Bagel, and Bagel-Thinking, we use textual descriptions of style images as a proxy due to the lack of multi-reference conditioning. However, these descriptions are often imprecise and fail to capture fine-grained stylistic attributes, leading to poor style consistency. In addition, irrelevant or verbose prompt content may interfere with content preservation and disrupt structural alignment. These results highlight the importance of multi-reference inputs for achieving faithful style transfer while maintaining content integrity.

Table 2: Quantitative comparison of style transfer methods across multiple metrics (**best** in bold, second-best underlined).

| Metric / Method | DeStyle2Style | OmniStyle | AD | StyleID | AesPANet | CSGO | StyleShot | STROTSS |
|---|---|---|---|---|---|---|---|---|
| DINO-Score ↑ | 0.8203 | 0.8606 | 0.8479 | **0.8828** | 0.8001 | 0.6714 | 0.6714 | 0.7677 |
| CLIP-Score ↑ | 0.2702 | **0.2777** | 0.2667 | 0.2731 | 0.2666 | 0.2370 | 0.1977 | 0.2544 |
| CSD-Score ↑ | **0.5606** | 0.5159 | 0.5256 | 0.4102 | 0.3019 | 0.5280 | 0.5276 | 0.4456 |
| Style Loss ↓ | **0.1170** | 0.1221 | 0.1322 | 0.1275 | 0.3455 | 0.1278 | 0.1288 | 0.1381 |
| Qwen-Content-Score ↑ | 8.1385 | 8.1277 | 7.8149 | **8.2283** | 7.9878 | 6.6793 | 4.6082 | 7.7821 |
| Qwen-Style-Score ↑ | **7.5763** | 7.4242 | 6.7531 | 6.5404 | 6.8722 | 7.0094 | 7.5445 | 6.9866 |
| Qwen-Aesthetic-Score ↑ | **8.7326** | 8.1681 | 7.9087 | 7.2955 | 7.1135 | 7.8304 | 8.1133 | 6.9987 |

Table 3: Quantitative comparison of image editing methods across multiple metrics (**best** in bold, second-best underlined).

| Metrics/Model | DeStyle2Style | USO | GPT-4o | Qwen-Image-Edit | FLUX-Kontext | Bagel | Bagel-Thinking |
|---|---|---|---|---|---|---|---|
| DINO-Score ↑ | 0.8203 | **0.8740** | 0.8506 | 0.7421 | 0.8132 | 0.7287 | 0.7183 |
| CLIP-Score ↑ | 0.2702 | 0.2681 | **0.2930** | 0.2375 | 0.2623 | 0.2320 | 0.2446 |
| CSD-Score ↑ | **0.5606** | 0.4441 | 0.5536 | 0.5576 | 0.5330 | 0.5494 | 0.5516 |
| Style Loss ↓ | 0.1170 | 0.1361 | **0.0380** | 0.1172 | 0.1499 | 0.1202 | 0.1204 |
| Qwen-Content-Score ↑ | 8.1385 | **9.0024** | 7.5388 | 7.1202 | 8.2676 | 7.7216 | 7.7355 |
| Qwen-Style-Score ↑ | 7.5763 | 4.6711 | **8.1156** | 7.2436 | 6.5395 | 6.6201 | 6.3715 |
| Qwen-Aesthetic-Score ↑ | 8.7326 | 9.2693 | 9.3507 | 9.5412 | 9.3351 | 9.3766 | **9.5980** |

Table 4: User study comparison between our method and representative style transfer approaches (**best** in bold, second-best underlined).

| Metric / Method | DeStyle2Style | OmniStyle | AD | StyleID | AesPANet | CSGO | StyleShot | STROTSS |
|---|---|---|---|---|---|---|---|---|
| Rank 1 (%) ↑ | **28.21** | 18.82 | 8.54 | 13.68 | 9.40 | 11.11 | 5.12 | 5.12 |
| Top 3 (%) ↑ | **58.95** | 56.40 | 25.62 | 38.46 | 35.75 | 35.88 | 20.49 | 28.45 |

## 4.4 USER STUDY

To complement the quantitative evaluation, we conducted a user study to assess the perceptual quality of stylized results. Participants were shown outputs from DeStyle2Style and other competing methods, and asked to rank their top three favorites based on: (1) *Style Preservation* — how well the style of the reference image is reflected; (2) *Content Preservation* — the degree to which structural details of the content image are retained; and (3) *Aesthetic Appeal* — overall visual quality. To reduce bias, image order was randomized and zooming was enabled. We collected 1,620 votes from

Table 5: User study comparison between our method and representative image editing methods (**best** in bold, second-best underlined).

| Metrics/Model | DeStyle2Style | USO | GPT-4o | Qwen-Image-Edit | FLUX-Kontext | Bagel | Bagel-Thinking |
|---|---|---|---|---|---|---|---|
| Rank 1 (%) ↑ | **34.56** | 8.64 | 32.72 | 12.96 | 5.55 | 2.49 | 3.08 |
| Top 3 (%) ↑ | 71.60 | 40.12 | **75.92** | 47.53 | 34.56 | 11.14 | 19.13 |

30 participants. As shown in Table 4 and Table 5, we report both Rank-1 proportions and Top-3 selection rates. The results show a clear preference for our method: it outperforms existing style transfer approaches (Table 2) and achieves performance close to GPT-4o (Table 3).

## 4.5 QUALITATIVE EVALUATION

**Comparison to Style Transfer Models.** As shown in Fig. 6, we qualitatively compare DeStyle2Style with several representative methods. Under the cartoon style (first row), others mainly apply color shifts, while DeStyle2Style generates clear cartoon-like characters, showing stronger stylization. Compared to optimization-based methods (AD, STROTSS), DeStyle2Style avoids content leakage, which often causes textures like trees to spill onto unrelated regions (bridges). DeStyle2Style also outperforms tuning-free models (OmniStyle, StyleShot, CSGO, Aes-PANET) by maintaining semantic consistency. It applies uniform styles to regions such as faces (second row) and bridges (last row), whereas others produce inconsistent textures and colors.

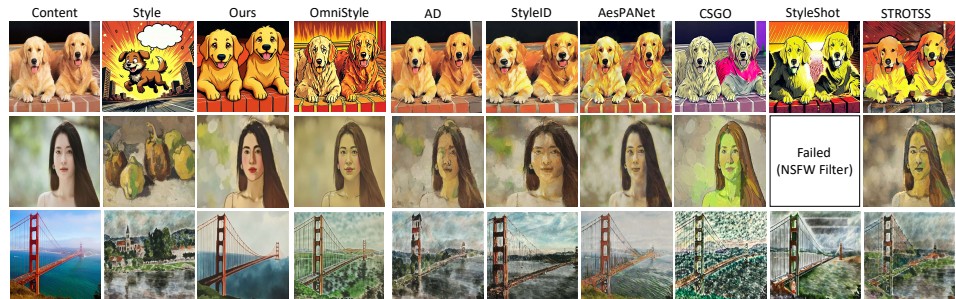

Figure 6: Qualitative comparison with other state-of-the-art methods. The missing result of StyleShot is filtered by its automatic NSFW detector.

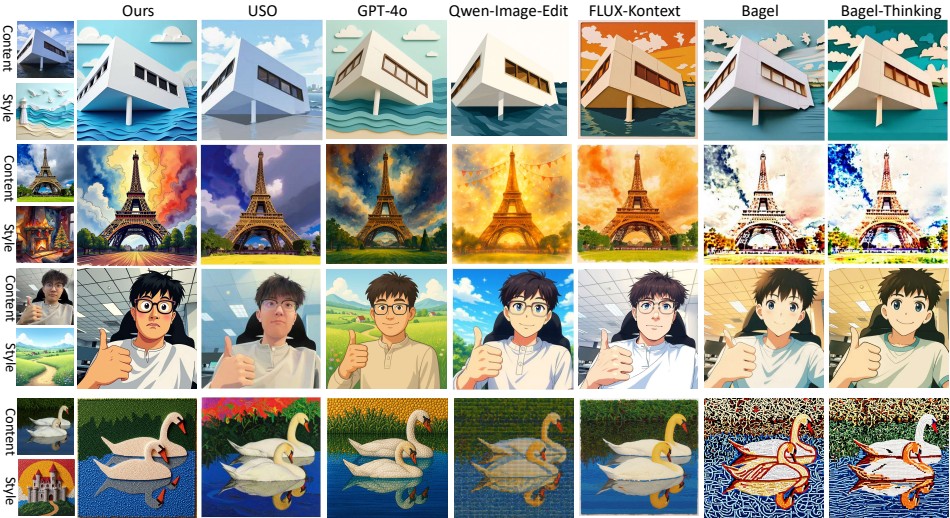

Figure 7: Comparison between our DeStyle2Style model and the existing image editing models.

**Comparison to the Image Editing Models.** Figure 7 presents a qualitative comparison between our method and several representative image editing models. We divide the analysis into two parts based on whether the model supports multi-image reference.

**(1) Comparison with GPT-4o and USO.** GPT-4o suffers from content leakage (e.g., Row 3) and noticeable color shifts, typically showing yellowish or overly warm tones compared to the refer-

ence style images (Rows 1–2), which compromise both content fidelity and style accuracy. USO maintains the structural integrity of the content image but exhibits insufficient stylization and fails to achieve faithful style transfer. In contrast, our method effectively preserves the content structure and accurately captures the intended style without introducing such artifacts.

**(2) Comparison with Open-Source Editing Models.** Since FLUX-Kontext, Qwen-Image-Edit, Bagel, and Bagel-Thinking do not support multi-image reference, we adopt a single-image input setup by converting the style image into a descriptive text instruction. However, these models struggle with complex style transfer tasks, such as the origami-inspired rendering in Row 1 or the pill mosaic in Row 4, and are generally limited to performing simple color adjustments. This limitation likely stems from the inherent difficulty of capturing complex visual styles through text descriptions alone. In contrast, DeStyle2Style leverages multi-image inputs to directly perceive and integrate visual style cues, enabling more faithful reproduction of stylistic elements.

## 5 CONCLUSION

We present DeStyle2Style, a novel framework that rethinks artistic style transfer as a data-centric problem. By introducing destylization as an inverse formulation, we address the long-standing challenge of lacking authentic supervision in style transfer tasks. Our proposed DeStyle-100K dataset provides high-quality training triplets constructed through destylization, enabling real artistic images, rather than synthetic outputs, to serve directly as supervision targets. This offers a more authentic supervision signal compared to prior pseudo-target approaches. Central to our pipeline are DestyleNet, a text-guided destylization model that reduces stylistic elements while preserving content, and DestyleCoT-Filter, a Chain-of-Thought-based quality assessment mechanism that enforces both content fidelity and style discrepancy. Furthermore, we introduce BCS-Bench, a benchmark with balanced stylistic diversity and content generality, enabling systematic evaluation of style transfer methods. Extensive experiments show that DeStyle2Style generates high-quality stylizations and consistently outperforms prior methods. Our work highlights that scalable and authentic supervision via destylization is essential for achieving reliable and faithful artistic style transfer.

## REPRODUCIBILITY STATEMENT

Dataset creation and processing steps are described in Section 3.3 and Appendix A.4. Implementation details are described in Sections 4.2 and Appendix A.4, including model architecture, training hyperparameters, and evaluation protocols. The code and dataset will be made publicly available in a future release.

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

## A  APPENDIX

We first discuss the limitations of our work and outline potential directions for future research (see Section A.1). We then present additional data samples from the DeStyle-100K dataset (see Section A.2). Next, we provide more stylization results of our method (see Section A.3). Finally, we give a detailed description of the dataset construction process. (see A.4).

### A.1  LIMITATIONS AND FUTURE WORK

As a data-driven approach, our method may lead to identity changes in stylized results due to noisy data. We will continue improving data quality by designing more robust filtering mechanisms and leveraging more diverse data to enrich the dataset. In addition, future work will explore caption-free destylization strategies to further enhance data generation quality.

### A.2  ADDITIONAL DATASET SAMPLES

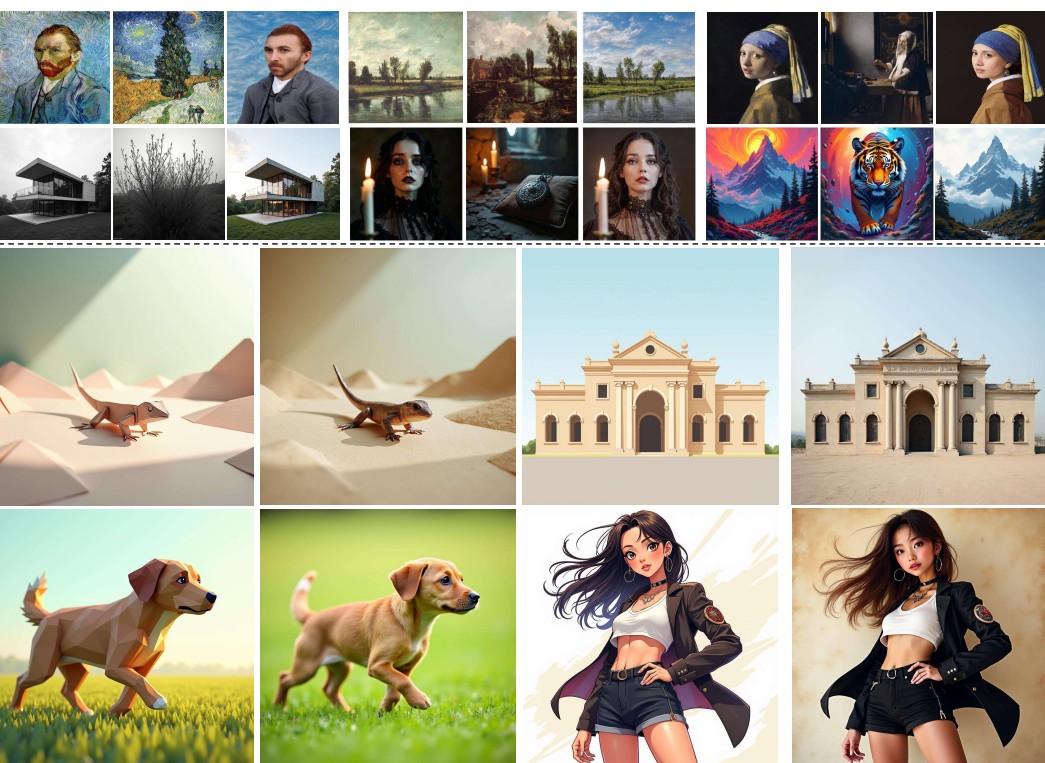

Figure 8: Top: Additional samples from DeStyle-100K. Each triplet (left to right) includes a style image, a reference image, and its destylized counterpart. Bottom: Destylization results by DestyleNet. Each pair (left to right) shows a style image and the corresponding destylized output.

As shown in the top part of Figure 8, we present additional samples from our DeStyle-100K dataset. The bottom part of Figure 8 illustrates more destylization results produced by our DestyleNet, including cases of origami, flat design, low-poly, and anime styles. Our method effectively preserves structural information while generating style-reduced, natural-looking content images.

### A.3  MORE RESULTS

#### A.3.1  MORE COMPARISONS WITH STYLE TRANSFER METHODS

As shown in Figure 9, we further compare our method with representative style transfer approaches. Optimization-based methods such as AD and STROTSS frequently suffer from content leakage, leading to noticeable distortions in the underlying content structures (see the 5th and 10th

columns). Methods including OmniStyle, StyleID, StyleShot, and CSGO exhibit insufficient stylization strength and often produce blurry appearances or disorganized textures. In contrast, our method achieves both strong and faithful stylization (e.g., photo-to-anime) and can handle more complex styles such as 3D origami. Our results also demonstrate noticeably higher image quality and aesthetic consistency compared to all baselines.

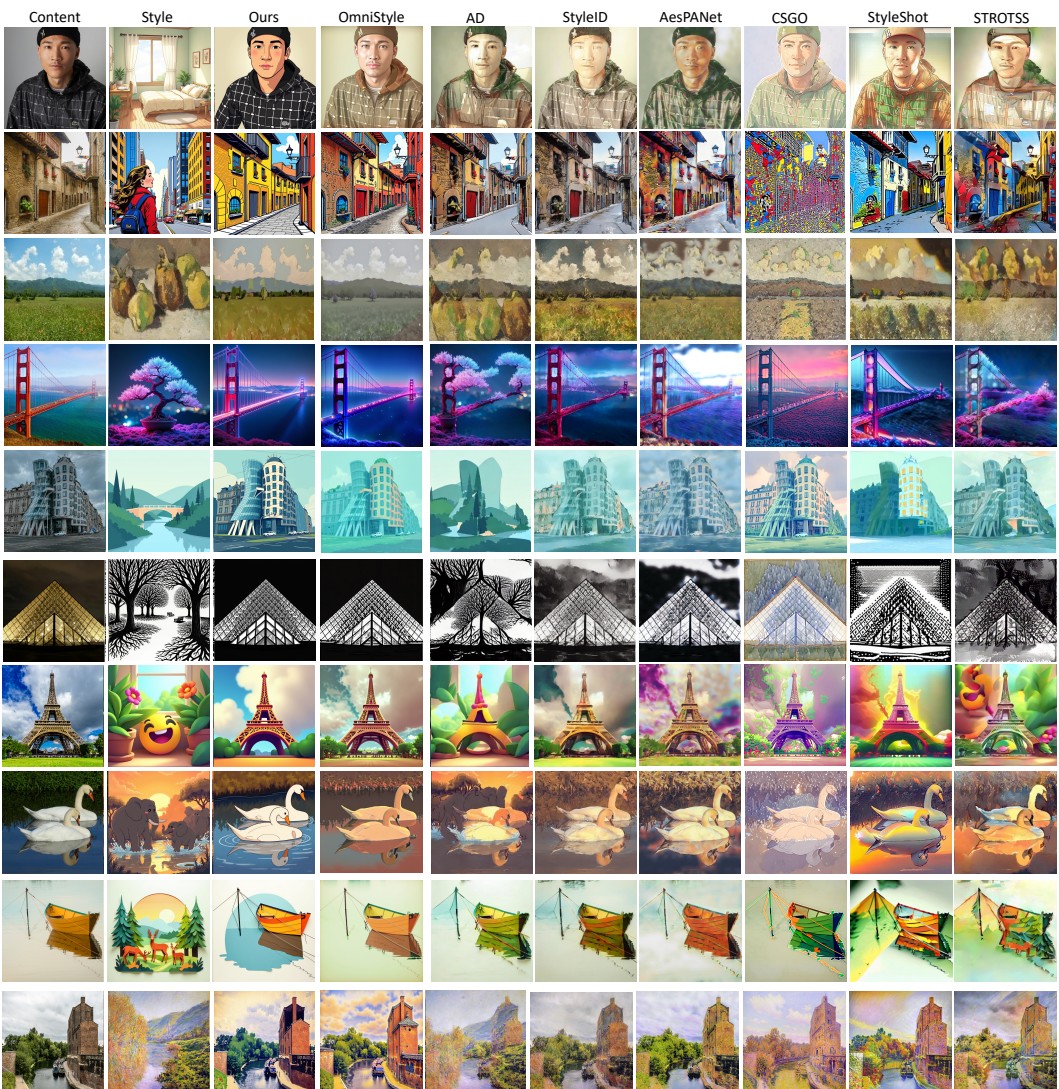

Figure 9: More comparisons of stylization results against other image style transfer models.

### A.3.2 MORE COMPARISONS WITH OPEN AND CLOSED-SOURCE IMAGE EDITING MODELS

As shown in Figure 10, we present further comparisons with image editing models. We observe that USO produces weaker stylization effects, while GPT-4o performs poorly in transferring real artistic styles (e.g., Row 1) and tends to suffer from semantic content leakage (e.g., Row 3). In contrast, our method achieves superior results. For FLUX-Kontext, Qwen-Image-Edit, Bagel, and Bagel-Thinking, the lack of multi-reference conditioning leads to relatively poor style consistency in their outputs.

### A.3.3 MORE RESULTS OF DESTYLE2STYLE

As shown in Figure 11, we present additional stylization results produced by our DeStyle2Style. The diverse style categories and high-quality details demonstrate the effectiveness of our approach.

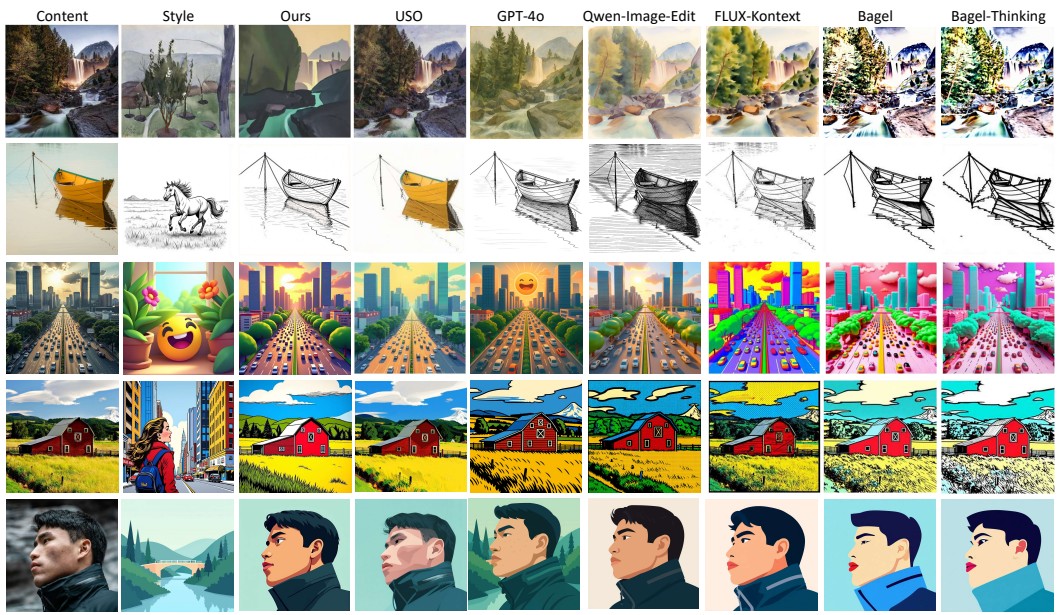

| Content | Style | Ours | USO | GPT-4o | Qwen-Image-Edit | FLUX-Kontext | Bagel | Bagel-Thinking |

Figure 10: More comparisons of stylization results against other image editing models.

Table 6: Quantitative evaluation of DeStyleNet's style reduced results.

| Test Set | ID Score | Style Removal Score | Image Quality | Image Aesthetic |
|---|---|---|---|---|
| Set 1 | 4.9467 | 4.0046 | 4.6123 | 4.5120 |
| Set 2 | 4.9397 | 3.9755 | 4.6123 | 4.5267 |
| Set 3 | 4.9190 | 3.9770 | 4.6229 | 4.5218 |
| Set 4 | 4.9314 | 3.9545 | 4.6202 | 4.5100 |
| Set 5 | 4.9358 | 3.9219 | 4.6193 | 4.5210 |
| Mean | 4.9345 | 3.9667 | 4.6174 | 4.5183 |

To further quantify the effectiveness of DeStyleNet, we conducted a comprehensive quantitative evaluation of its de-stylization results, as shown in Table 6. Specifically, we evaluated the de-stylization results by randomly selecting 1,000 samples at a time, repeating this process across five separate trials. To ensure a thorough assessment, we designed four evaluation metrics: **ID Score**, which measures the identity consistency of the de-stylized images; **Style Removal Score**, which quantifies the degree to which style information is removed; **Image Quality**, which evaluates the overall quality of the de-stylized images; and **Image Aesthetic**, which reflects the aesthetic appeal of the resulting images.

For scoring, we employed QwenVL-Max, utilizing carefully designed prompts for each metric. The scoring range for all metrics was standardized to 0–5, where, for instance, an ID Score of 0 indicates entirely inconsistent identities, while a score of 5 denotes complete consistency.

As demonstrated in Table 6, DeStyleNet consistently achieves high-quality de-stylization results. Specifically, it preserves identity information with remarkable fidelity (mean ID Score of 4.9345) while demonstrating effective style removal (mean Style Removal Score of 3.9667). Furthermore, the de-stylized images exhibit high levels of image quality score (4.6174) and aesthetics score (4.5183). These results collectively validate the effectiveness of DeStyleNet in achieving de-stylization while maintaining both identity consistency and image quality.

### A.3.4 IMPACT OF BACKBONE MODEL SIZE

To investigate the effect of model scale on style transfer performance, we compare SD3-Medium (2B parameters) with Flux-Dev (12B parameters) fine-tuned on our DeStyle-100K dataset. The quantitative results are presented in Table 7.

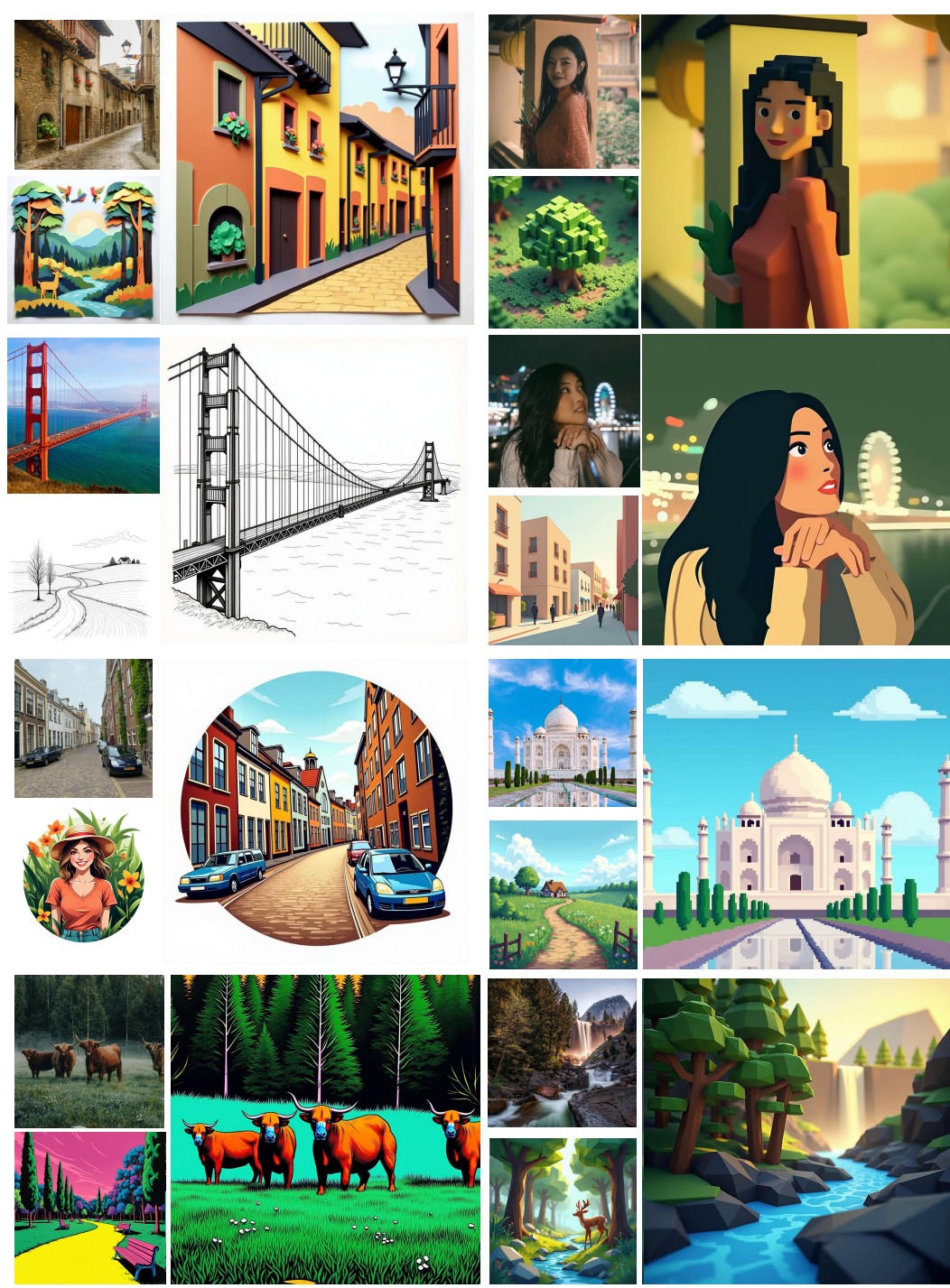

Figure 11: More stylization results of DeStyle2Style.

**Quantitative Analysis.** From Table 7, we observe that the larger Flux-Dev model demonstrates advantages in several metrics: it achieves higher DINO-Score (0.8203 vs 0.7473) and CLIP-Score (0.2702 vs 0.2356), indicating better semantic feature preservation and text-image alignment. On the other hand, the smaller SD3-Medium model excels in Style Loss (0.0518 vs 0.1170), Qwen-Content-Score (8.4413 vs 8.1385), and Qwen-Aesthetic-Score (9.2032 vs 8.7326).

Notably, despite having 6× fewer parameters, SD3-Medium achieves comparable or even better performance on the core style transfer metrics: CSD-Score (0.5341 vs 0.5606), Style Loss (0.0518 vs 0.1170) and Qwen-Style-Score (7.4789 vs 7.5763). This suggests that our DeStyle-100K dataset

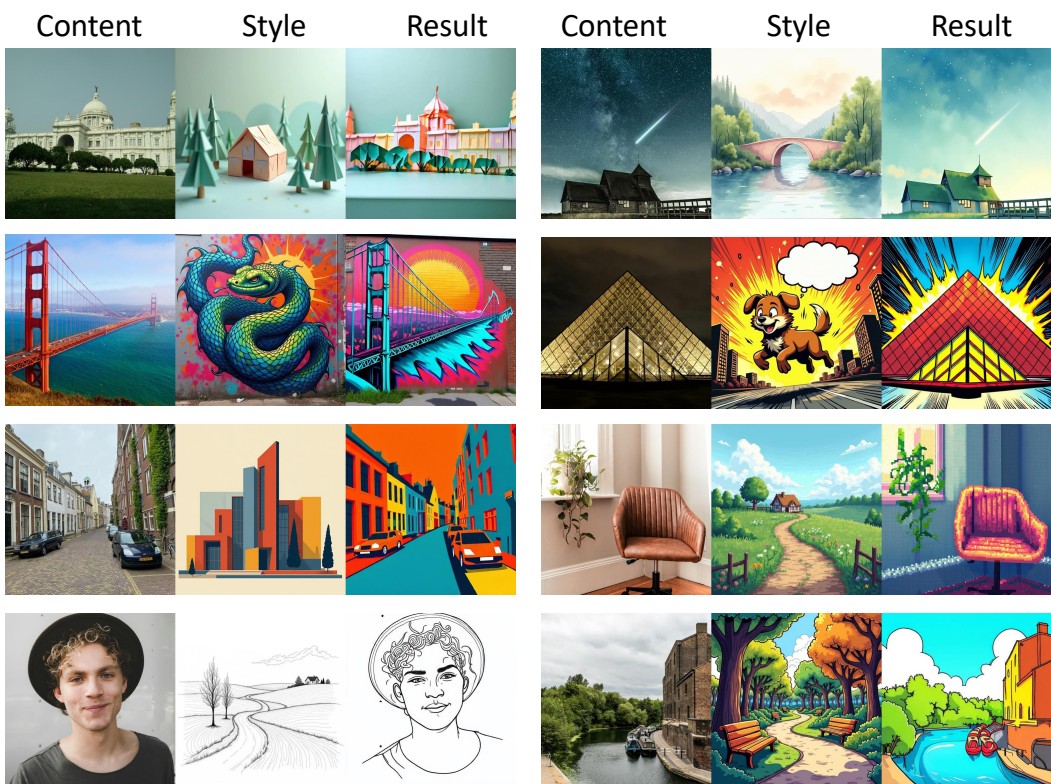

Figure 12: Style transfer results of stable-diffusion-3-medium (2B) fine-tuned on our DeStyle-100K dataset.

enables effective style transfer training even with significantly smaller models, without substantial degradation in style quality.

**Qualitative Analysis.** Figure 12 presents SD3-Medium's style transfer results across diverse content and style combinations. The model successfully transfers various artistic styles including cartoon animation, geometric abstraction, pixel art, line art and 3D Papercraft, while preserving the semantic content of the original images. The results demonstrate high-quality stylization with vibrant colors, clear structural details, and faithful style representation, confirming the quantitative findings.

The above quantitative and qualitative results further validate the effectiveness of our DeStyle-100K dataset and training pipeline, demonstrating their architecture-agnostic reliability. Notably, SD3-Medium with only 2B parameters achieves competitive style transfer performance compared to the 12B Flux-Dev model, confirming that our approach generalizes effectively across different model scales and architectures.

Table 7: Quantitative comparison of different backbone finetuned on our DeStyle-100K dataset.

| Backbone | Parameters Size | DINO Score | CLIP Score | CSD Score | Style Loss | Content | Qwen Style | Aesthetic |
|---|---|---|---|---|---|---|---|---|
| Flux-Dev | 12B | **0.8203** | **0.2702** | **0.5606** | 0.1170 | 8.1385 | **7.5763** | 8.7326 |
| SD3-Medium | 2B | 0.7473 | 0.2356 | 0.5341 | **0.0518** | **8.4413** | 7.4789 | **9.2032** |

## A.4 ADDITIONAL DETAILS ON DATASET CONSTRUCTION

In this section, we provide detailed information on the dataset construction process. Specifically, Section A.4.1 describes the collection of real artistic images, Section A.4.2 explains the synthesis of

Table 8: Construction of a content tree comprising six major categories, including Human, Scene, Architecture, Object, Animal, and Plant, each with ten fine-grained subcategories. This hierarchical taxonomy serves as the content basis for generating style images.

| Category | Fine-grained Subcategories |
|---|---|
| Human | Single portrait (face close-up), Half-body (upper body), Full-body (standing), Two people (interaction or pose), Group of people (3–5 individuals), Child (toddler or school age), Elderly person, Person in traditional clothing, Fantasy character, Professional (e.g., doctor) |
| Scene | Urban street (with buildings and people), Modern cityscape (skyscrapers, skyline), Indoor room (bedroom, kitchen, office), Park (trees, paths, benches), Countryside (fields, rural roads), Mountain landscape, Forest scene, Beach or coast, Night city scene, Fantasy or magical landscape |
| Architecture | Modern house or villa, Apartment building, Traditional Asian architecture, Classical European building, Futuristic building, Cottage or cabin, Bridge, Skyscraper, Church or mosque, Historic ruin or monument |
| Object | Chair or sofa, Table or desk, Laptop or smartphone, Camera, Musical instrument (e.g., guitar), Vehicle (car, bicycle, motorcycle), Book, Backpack or bag, Watch or jewelry, Toy (e.g., teddy bear) |
| Animal | Dog, Cat, Horse, Bird (e.g., parrot, owl), Fish (e.g., goldfish, clownfish), Lion or tiger, Elephant, Butterfly, Snake or lizard, Fantasy creature (e.g., dragon) |
| Plant | Flower (e.g., rose, sunflower), Tree (e.g., pine, cherry blossom), Potted plant (e.g., monstera, cactus), Bush or shrub, Field of flowers, Bonsai tree, Grass or lawn, Hanging plant or vine, Tropical plant, Forest vegetation |

Table 9: We define 65 mainstream artistic styles for synthesizing style images.

Anime, Art Nouveau, Bauhaus, Chalk Art, Chinese Traditional, Comic Art, Constructivism, Cubism, Cyberpunk, Dark Fantasy, Etching, Expressionism, Fantasy, Fauvism, Fresco, Futurism, Gothic Horror, Graffiti, Ink Wash, Japanese Ukiyo, Line Art, Linocut, Lithography, Low Poly, Manga, Pastel, Persian Miniature, Pixel Art, Pointillism, Pop Art, Screen Printing, Stained Glass, Stencil, Sumi-e, Synthwave, Tattoo Art, Vaporwave, Voxel Art, Watercolor, Weirdcore, Woodcut, Oil Painting, Pencil Sketch, Charcoal Drawing, Ukiyo-e, Chinese Ink Painting, Western Classical Painting, Impressionism, Abstract Art, Cartoon, Ghibli-style, American Comic, Children's Book Illustration, Hand-drawn Illustration, Flat Design, Origami, 3D Render, Steampunk, App UI, Photorealistic, Magic Realism, Minimalist, Black and White, Film Look, Surrealism, Neon, African Tribal

style images, Section A.4.3 outlines the filtering and quality control procedures, and Section A.4.4 presents statistics and visualizations of the data set.

### A.4.1 COLLECTION OF REAL ARTISTIC IMAGES

As shown in Figure 13 (a), we begin by collecting real-world artworks from two publicly available datasets: WikiArt Tan et al. (2019) and the National Gallery of Art National Gallery of Art (2025). Each image is accompanied by metadata such as artist and art movement. We use these annotations to define fine-grained style categories in the form of *artist–movement* pairs (e.g., Van Gogh–Post Impressionism), resulting in 2,641 unique style categories. For each category, we calculate the CLIP-based image-text similarity between each artwork and its corresponding label. Images with similarity scores below the category average are discarded, yielding an initial subset of 58K candidate images. We further refine this subset by removing blurry images, low-resolution samples, non-artworks, and duplicates. During manual inspection, we observed that many artistic images were overly abstract and lacked clearly interpretable semantic content, making them unsuitable for the destylization task. To address this, we employ InternVL2.5-7B Chen et al. (2024) to perform semantic content analysis, focusing on six primary content types: **human, object, animal, plant, scene, and architecture**. Images not associated with any of these categories are filtered out. Figure 14 illustrates examples of excluded abstract images. After this multi-stage filtering process, we retain a final set of 10K high-quality artistic images, encompassing works by 669 artists (e.g., Van Gogh and Monet) and spanning 117 distinct art movements (e.g., Impressionism, Baroque). To ensure visual consistency, all images are resized to $1024 \times 1024$ pixels.

### A.4.2 GENERATION OF STYLE IMAGES

To further enhance the diversity of our dataset, we synthesize a large number of stylized images via generative modeling. We begin by constructing a comprehensive content taxonomy (Table 8), which consists of six major categories, each encompassing ten fine-grained subcategories. For example, the Human category includes subclasses such as single portrait, half-body, full-body, and multi-person interaction. In addition, we define 65 mainstream digital art styles, including Anime, Line Art, and Watercolor and more, as listed in Table 9.

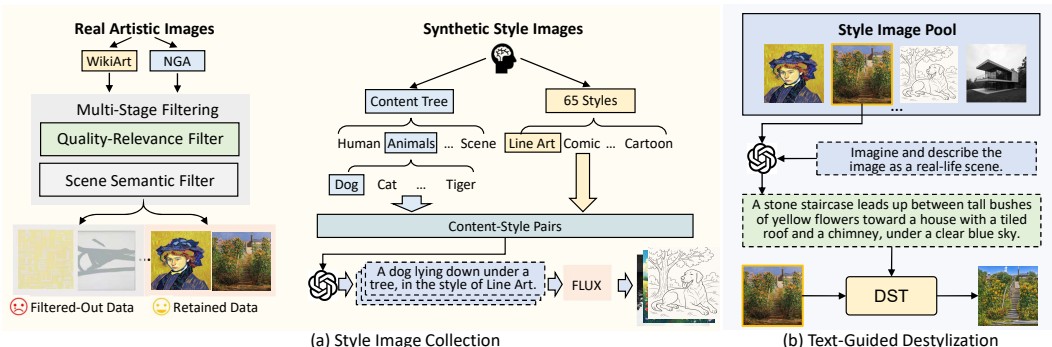

(a) Style Image Collection        (b) Text-Guided Destylization

Figure 13: (a) Style image collection and (b) text-guided destylization pipeline.

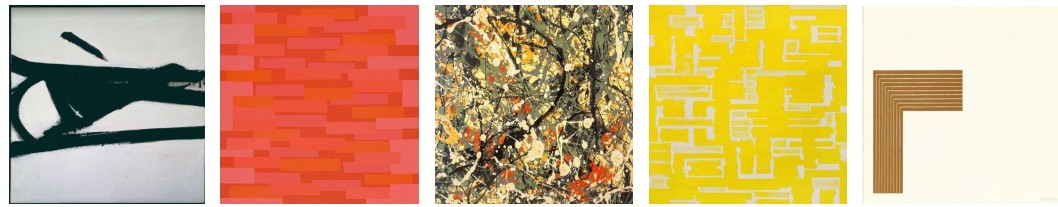

Figure 14: Examples of real artistic images that were excluded due to their overly abstract nature and lack of clearly interpretable semantic content, which makes them unsuitable for the destylization task.

### A.4.3 FILTERING AND QUALITY CONTROL

As shown in Figure 15, we utilize GPT-4o to filter low-quality style-desty image pairs, assessing their quality from two key perspectives: content preservation and style discrepancy. As described in the main text, we adopt a fine-grained, multi-stage assessment strategy based on Chain-of-Thought reasoning. Figures 16 and 17 show the prompt templates used for the two evaluation tasks.

For content preservation, GPT-4o is first instructed to identify all key semantic regions and objects in the style (left) image. Then, for each identified region, it evaluates whether the corresponding content is faithfully preserved in the destylized (right) image. The final score is computed by aggregating the evaluations of all key regions. To ensure scoring consistency, we define a detailed scoring criterion summarized below:

- **5**: All objects and regions are perfectly preserved with no perceptible errors.
- **4**: Nearly perfect; all objects are present and clearly reconstructed, with only extremely minor, barely visible issues.
- **3**: At least one object or region is slightly degraded or inaccurately rendered (e.g., blurry, simplified, off-shape).
- **2**: Multiple objects show errors or degradation; several elements are not well-preserved.
- **1**: Major objects are missing, malformed, or hallucinated.
- **0**: Most content is lost or the scene is unrecognizable.

The evaluation strictly focuses on the preservation of semantic content. Style-related differences (e.g., color, brushstroke, artistic texture) must be ignored. If any meaningful object or region from the left image is not properly preserved in the right image, the score should be reduced accordingly. A similar multi-stage, fine-grained reasoning process is applied for the assessment of style discrepancy.

### A.4.4 DATASET STATISTICS AND VISUALIZATIONS

As shown in Figure 18, we visualize the distribution of synthesized stylized images. The left plot shows a balanced coverage of six content categories: Animal, Human, Scene, Plant, Object, and Architecture. The right plot shows an even distribution across 65 styles, which helps mitigate long-tail

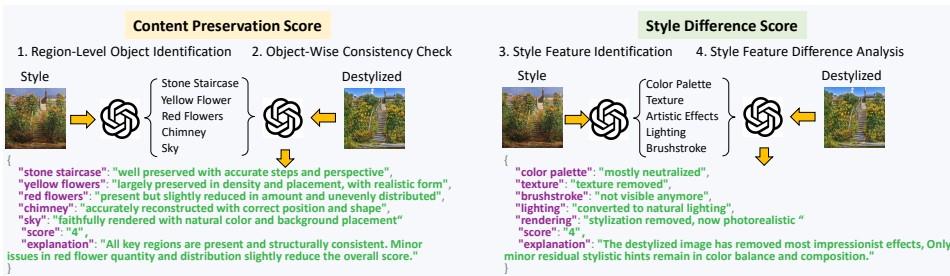

Figure 15: The pipeline of DestyleCoT-Filter. DestyleCoT-Filter assesses each ⟨ style, destylized ⟩ pair from two aspects: content preservation and style discrepancy, using GPT-4o with region-level and attribute-level Chain-of-Thought reasoning.

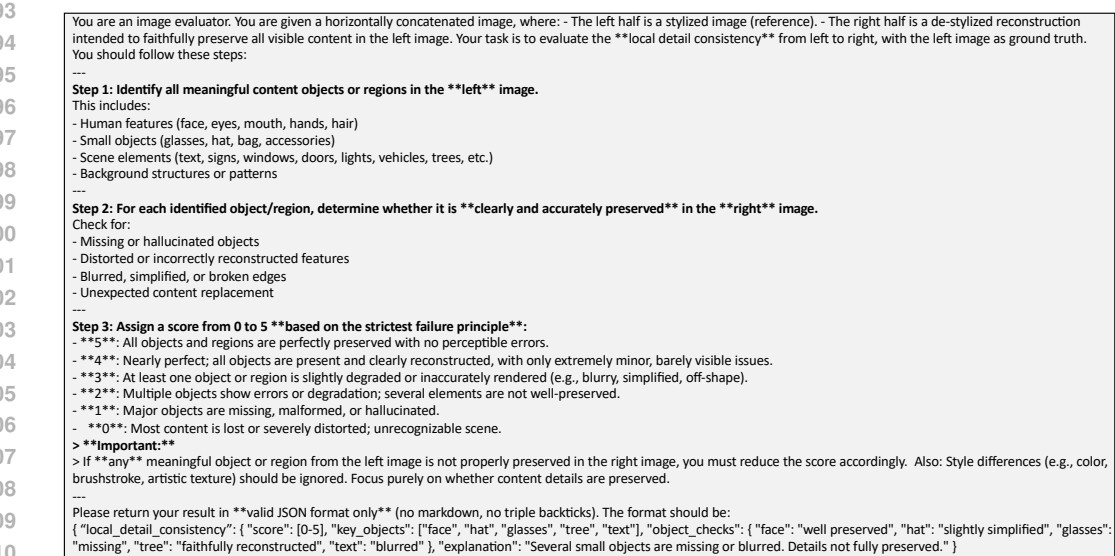

Figure 16: Text prompt used by DestyleCoT-Filter for content preservation assessment.

effects from data imbalance. As shown in Table 10, we summarize 117 real-world artistic movements based on authentic artworks. Due to the large number of associated artists, we omit the full list of artist names.

LARGE LANGUAGE MODEL (LLM) USAGE

Parts of the manuscript were polished for grammar and style using LLM under the authors' direction. The authors verified and edited all generated text, and the model was not involved in generating research ideas, experimental design, or results.

You are an image evaluator. You are given a horizontally concatenated image, where: The **left** half is a stylized reference image. The **right** half is a de-stylized reconstruction. Your task is to evaluate the **style difference** between the two halves. Focus only on **stylistic aspects** — do **not** consider object preservation or semantic content.

---
**Step 1: Observe all stylistic features in the left image.**

This includes:
- Color tones, saturation, and palettes
- Texture characteristics (e.g., smooth, rough, brush-like, paper-like)
- Artistic effects (e.g., oil painting, watercolor, sketch, cartoon, photorealism)
- Lighting style, shading, shadows
- Rendering irregularities or stylization patterns
---
**Step 2: Compare these style features with the right image.**
Identify and describe:
- Which stylistic elements were **removed, softened, or preserved**
- Whether the right image has become **neutralized**, **photorealistic**, or **completely different**
- Whether any **stylization patterns** are still visible
---
**Step 3: Assign a score from 0 to 5 based on **how much the style has changed** from left to right:**
- **5**: Completely different styles; all stylization removed or transformed. The right image looks natural or neutral.
- **4**: Most stylistic features removed, only faint traces remain (e.g., slight texture or lighting retained).
- **3**: Mixed: some styles clearly removed, but some textures or colors are still similar.
- **2**: Many stylistic features still remain; only partial de-stylization achieved.
- **1**: Only very subtle changes; most stylization patterns are still present.
- **0**: No visible difference in style between the two images.
> ⚠️ **Important:** Ignore all content differences. Only judge **visual style and artistic appearance**.

Figure 17: Text prompt used by DestyleCoT-Filter for style discrepancy assessment.

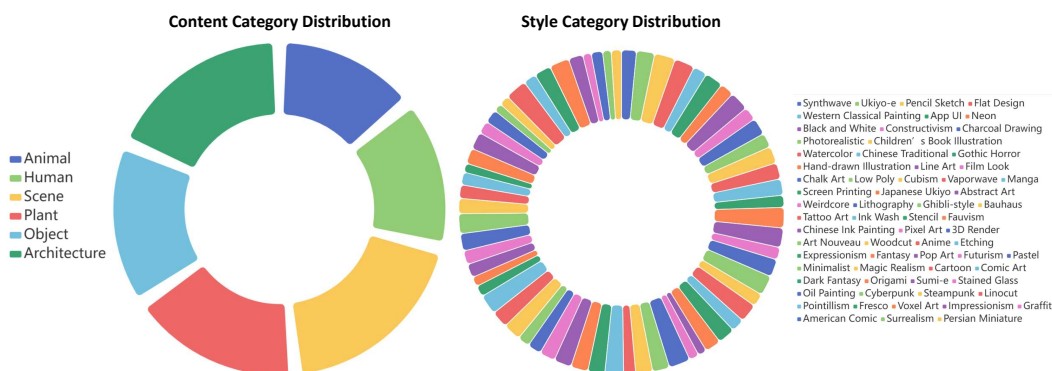

Figure 18: The approximate distribution of synthesized style images across content and style categories.

Table 10: The DeStyle-100K dataset encompasses 117 real-world artistic movements.

Etching on laid paper, Nature print, Charcoal on tan paper, Pop Art, Pen and black ink and watercolor, Red chalk, Mannerism Late Renaissance, Engraving and aquatint in black on wove paper, Palladium print, New Realism, Dye imbibition print, Ukiyo-e, Watercolor, graphite, and gouache on paperboard, Watercolor, graphite, and gouache on paper, Graphite and watercolor, Pen and ink and gouache on paper, Photogravure on beige thin slightly textured paper, Etching and aquatint in black on wove paper, Watercolor and graphite on paper, Etching and drypoint on Japanese paper, Watercolor, Gelatin silver print on aluminum, Watercolor, gouache, and graphite on paperboard, Color photogravure on wove Somerset Satin White paper, Watercolor, colored pencil, and graphite on paper, Bronze, Photogravure on cream wove paper, Romanticism, Nicolas Chifflart, Pen and black ink on wove paper, Hand printed wood engraving on Japanese paper, Gelatin silver print mounted on paperboard, Early Renaissance, Expressionism, Watercolor over graphite on wove paper, Etching heightened with white on blue laid paper, Dye diffusion transfer print (Polacolor), Chromogenic print, Beaulieu, Black and white photograph, Watercolor and graphite on wove paper, Autochrome, Etching and drypoint in black on wove paper, Charcoal on laid paper, Realism, Etching and engraving on laid paper, Watercolor and graphite on paperboard, Watercolor, gouache, and graphite on paperboard, Impressionism, Gelatin silver print, Engraving on laid paper, Art Nouveau Modern, Symbolism, Marie Vien, Mémin, Watercolor and graphite, Baroque, Cyanotype, Northern Renaissance, Daguerreotype, Analytical Cubism, Post Impressionism, Wood engraving in black on laid paper, Fauvism, Pen and brown ink with gray wash on paper, Etching with engraving and drypoint, Watercolor, colored pencil, and graphite, Graphite on wove paper, Charcoal and graphite, Graphite, Graphite on paper, Louis Forain, Pen and black ink with brown wash on paper, Soot on found paper, Oil on cardboard, Watercolor, colored pencil, and graphite on paper, Watercolor and graphite on laid paper, Etching with drypoint and roulette on chine collé, Engraving on laid paper, Lautrec, Etching on wove paper, Gelatin silver print mounted on tissue paper, Baptiste, Contemporary Realism, Etching and drypoint in brown on laid paper, Platinum print, Latour, Salted paper print, Bresson, Pen and brown ink and watercolor on laid paper, Pen and black ink with gray wash on paper, Pointillism, Pen and brown ink with brown wash on laid paper, Albumen print, Bottom panel of two parts: pastel, charcoal, wax crayon, Graphite on laid paper, Rococo, Watercolor on wove paper, Dornburg, Cubism, Oil on canvas, Naive Art Primitivism, Watercolor, colored pencil, pen and ink, and graphite, Watercolor on laid paper, Color Field Painting, Louis Barye, Lithograph in black on wove paper, Pen and brown ink and watercolor, Platinum print mounted on laid paper, Lithograph on wove paper, High Renaissance, Denis Baldus, Fresco, Lithograph in black on Arches 88 wove paper, Pen and ink on paper, Mezzotint [progress proof], Oil on wood

