# OpenReview forum: "DeStyle2Style: Scalable Destylization-Driven Data Generation for Artistic Style Transfer"
_ICLR.cc/2026/Conference — Submitted to ICLR 2026_

### Official Review · Reviewer_DF8q · 2025-10-17

**Soundness:** 2
**Presentation:** 3
**Contribution:** 2
**Rating:** 6
**Confidence:** 4

**Summary:**

This paper reframes artistic style transfer as a data generation problem. First, the authors develop DestyleNet—a text-guided destylization model that reconstructs natural images with reduced artistic styles. Second, they leverage GPT-4o (equipped with a Chain-of-Thought strategy) to build DestyleCoT-Filter, which enables the creation of DeStyle-100K, a new high-quality dataset. Third, using this dataset for training, they propose DeStyle2Style (based on FLUX-Dev), which achieves style transfer results comparable to SOTA approaches. Finally, they introduce BCS-Bench, a benchmark featuring balanced stylistic diversity and content generality, designed for the systematic evaluation of style transfer methods.

**Strengths:**

+ Using destylization offers a new perspective for creating paired data for artistic style transfer, enabling more authentic and higher-quality supervision signals.

+ The high-quality DeStyle-100K dataset and BCS-Bench benchmark (if publicly released) would provide valuable resources for the research community.

+ Both qualitative and quantitative comparisons validate the effectiveness of the constructed dataset and the DeStyle2Style model.

**Weaknesses:**

- DestyleNet is trained via full fine-tuning on a mere 60K paired samples—data synthesized using existing style transfer methods. As the authors highlight in L358, destylization is inherently a challenging task; this limited training data may undermine DestyleNet’s robustness. A more critical concern arises from its training paradigm: relying on outputs of existing style transfer methods means DestyleNet essentially learns the average of these methods’ inverse processes. This leads to a question: how can biases inherent to the base style transfer methods be avoided? For example, some methods prioritize color transfer over other stylistic elements, while others tend to distort content—would DestyleNet then only mimic such behaviors (e.g., merely removing colors or altering content) instead of achieving true, comprehensive destylization? Compounding these issues, the paper also lacks clarity on how to measure and evaluate DestyleNet’s effectiveness: no quantitative results or relevant discussions are provided to validate whether DestyleNet actually fulfills its intended destylization purpose.

- The primary novelty of this paper likely lies in its destylization data construction pipeline and the resulting dataset. In contrast, the architectures of DestyleNet and DeStyle2Style are largely inherited from existing mainstream frameworks such as OmniControl and Flux-Kontext.

**Questions:**

Please see weaknesses above.

---

> ### Author Response · Authors · 2025-11-19
>
> Thank you for your thoughtful review. We sincerely appreciate your recognition of **our destylization-based approach as a new perspective for paired data construction, the potential value of the DeStyle-100K dataset and BCS-Bench, and the effectiveness of our method demonstrated through both qualitative and quantitative comparisons**. Below, we address each of your concerns individually.
>
> > Response to W1.
>
> Thank you for the thoughtful and detailed feedback.
>
> * We fully agree that destylization is inherently a challenging task, and DestyleNet may not always produce perfect outputs.
> To address this, we introduce **DestyleCoT-Filter, a multi-stage MLLM-based filtering module that evaluates each destylized result along two dimensions, content preservation and style discrepancy, and removes low-quality samples**. This ensures that only high-quality data are used in the final training of the style transfer model.
> * As demonstrated in Table 3 and Table 5, **our model achieves performance comparable to strong closed-source models like GPT-4o**, which indirectly validates the **quality and effectiveness of our destylization process**.
> * Additionally, **DestyleNet is capable of handling a wide variety of style types, rather than merely removing color or texture**. As shown in Figure 2 (main paper) and Figure 8 (appendix), it successfully destylizes both **minimal styles (e.g., grayscale, line art) and complex styles such as 3D rendering, papercraft, and flat design**. These results demonstrate that **DestyleNet achieves effective and reliable destylization across diverse artistic styles**.
>
>
> > Response to W2.
>
> Thank you for the valuable comment.
> * As **our main contribution lies in the destylization-based data construction pipeline and the resulting DeStyle-100K dataset**, we intentionally chose to **use a simple and unmodified architecture (FLUX-Dev) to better isolate and demonstrate the effectiveness of our data**. This design choice helps us focus on validating the impact of data quality, **rather than introducing additional architectural variables**.
> * Notably, even with this basic setup, **our method achieves strong results across both quantitative and qualitative evaluations**, which supports the value of our proposed data-centric approach.
> * We would like to highlight that **our dataset and pipeline are model-agnostic**, which allows them to **be readily applied to fine-tune various backbones**, such as SD3-Medium-2B. For further details, we kindly refer you to **Figure 12, Table 6, and Section A.3.4 in the Appendix**. We hope that our contribution can inspire future works to explore and build on our data pipeline by leveraging stronger architectures.

---

> > ### Comment · Reviewer_DF8q · 2025-11-26
> >
> > Thanks for the rebuttal. I think this paper has made a good first step in applying destylization to paired data construction. However, since the destylization data construction pipeline is the main contribution, more in-depth analyses and discussions of destylization are required—especially regarding the evaluation of destylization effectiveness and the avoidance of biases inherent to base style transfer methods in destylization.

---

> > > ### Author Response · Authors · 2025-11-27
> > >
> > > Thank you very much for your constructive feedback. **We completely agree that a deeper analysis of the destylization process is essential to support our main contribution**. In response, we have **added a new quantitative evaluation of DeStyleNet’s destylization results, now presented in Table 6 (line 830-858) of the revised paper**.
> > >
> > > Specifically, we conducted **5 independent trials, each randomly sampling 1,000 destylized images, and evaluated them using 4 comprehensive metrics**:
> > >
> > > * **ID Score**: Measures identity consistency between the destylized image and its original content image.
> > > * **Style Removal Score**: Quantifies the extent to which style has been effectively removed.
> > > * **Image Quality**: Assesses the image quality of the destylized output.
> > > * **Image Aesthetic**: Reflects the overall aesthetic appeal of the destylized result.
> > >
> > > All metrics were computed using QwenVL-Max, prompted appropriately for each evaluation dimension. The scoring range for all metrics is standardized to **0–5, where higher values indicate better performance**.
> > >
> > >
> > > | Test Set | ID Score | Style Removal Score | Image Quality | Image Aesthetic |
> > > |----------|----------|---------------------|---------------|------------------|
> > > | Set 1    | 4.9467   | 4.0046              | 4.6123        | 4.5120           |
> > > | Set 2    | 4.9397   | 3.9755              | 4.6123        | 4.5267           |
> > > | Set 3    | 4.9190   | 3.9770              | 4.6229        | 4.5218           |
> > > | Set 4    | 4.9314   | 3.9545              | 4.6202        | 4.5100           |
> > > | Set 5    | 4.9358   | 3.9219              | 4.6193        | 4.5210           |
> > > | **Mean** | **4.9345** | **3.9667**         | **4.6174**    | **4.5183**       |
> > >
> > >
> > > Across 5 independent trials (each with 1,000 randomly sampled destylized images), our method achieved a **mean ID Score of 4.9345, a Style Removal Score of 3.9667, an Image Quality score of 4.6174, and an Image Aesthetic score of 4.5183**. These results demonstrate that **our method consistently achieves effective style removal while faithfully preserving identity and delivering high-quality outputs—supporting the effectiveness and generalizability of our destylization-based data construction pipeline.**
> > >
> > > We sincerely appreciate your insightful suggestion, which directly helped us improve the rigor and clarity of our evaluation.
> > >
> > > **If there are any remaining concerns or additional questions, we are more than willing to address them in full.**

---

> > > > ### Author Response · Authors · 2025-11-28
> > > >
> > > > Dear Reviewer,
> > > >
> > > > Thank you again for your valuable time and feedback. We truly appreciate your earlier comments and suggestions. We would like to kindly check whether our previous response has adequately addressed your concerns.
> > > >
> > > > If there are any remaining questions or issues, we are more than willing to provide further clarification or evidence. We sincerely look forward to your continued feedback and to further constructive discussion.

---

### Official Review · Reviewer_NvXS · 2025-10-17

**Soundness:** 2
**Presentation:** 2
**Contribution:** 2
**Rating:** 6
**Confidence:** 5

**Summary:**

This paper presents DeStyle-100K, a large-scale, high-quality dataset for supervised image style transfer built via a novel destylization pipeline. Using a DestyleNet model to reverse stylization and a DestyleCoT-Filter for quality control, it provides 100K aligned triplets <de-stylized image, reference image, style image>. Trained on this dataset, FLUX achieves superior results, reframing style transfer as a data-centric task.

**Strengths:**

1.The proposed dataset includes 100K high-quality triplet data points, which is highly beneficial to the style transfer community.

2.The complete construction of the pipeline is advantageous to the community.

3.The visual results of DestyleNet and the trained FLUX are good, content images are in the real-world domain and the stylized results are satisfactory.

4.Compared to the similar work OmniStyle[1], this paper have more diverse style images, help model to have better generalization ability.

[1]OmniStyle: Filtering High Quality Style Transfer Data at Scale, CVPR 2025.

**Weaknesses:**

1.The DeStyle model is trained on a dataset processed by current image-driven style transfer models. The upper bound of these methods also represents the limitation of the DeStyle-100K dataset. How to solve it?

2.As mentioned in [1], the de-stylized image and reference image have the consistent structure, is this will result in limited domain for image-driven style transfer? For example for Abstract style, if so, how to solve it?

3.In your triplets, the style image are constructed from style similarity based on CSD model. For my opinion, CSD model is not reliable because it is trained on WIKIART (most style are oil paintings). Picking from this metric might cause the different style between reference image and style image.

4.Does the proposed DestyleNet perform better than the destyle model in USO[1] that build on a powerful customization framwork UNO? Your proposed data curation pipeline is similar to USO.

[1]USO: Unified Style and Subject-Driven Generation via Disentangled and Reward Learning

If the authors solve all my concerns, I'd love to raise my score.

**Questions:**

1. Could you provide more visual results compared to SOTA style transfer methods? Only 3 samples in Figure 6.

2. Could you provide the comparison with SOTA methods AlignedGen[1].

[1]AlignedGen: Aligning Style Across Generated Images, NIPS 2025.

---

> ### Author Response · Authors · 2025-11-19
>
> Thank you for your thoughtful review. We sincerely appreciate **your recognition of our 100K high-quality triplet data points, the complete construction of our pipeline, the good visual results of DestyleNet and the trained FLUX model, and the use of more diverse style images compared to OmniStyle, which may help improve generalization ability**. Below, we address each of your concerns individually.
>
> > Response to W1.
>
> Thank you for pointing this out—we fully agree with your observation.
> * Prior works such as **OmniStyle and CSGO construct their datasets by applying existing style transfer models to synthesize stylized targets**. However, such stylized images often **suffer from visual artifacts, which limits the quality and reliability of the resulting supervision**.
> * This is precisely why our approach does not treat synthetic stylized images as supervision targets. Instead, **we use the original, real style image as the sole supervision target, providing an authentic and high-fidelity supervision signal**. **The destylized image generated by DestyleNet is only used as the content input, not as ground truth**. **While the destylization process may not be perfect, small imperfections introduce beneficial content variations that act as data augmentation, improving robustness during training**.
> * This design ensures that the supervision in DeStyle-100K remains reliable. As demonstrated in our experiments, **fine-tuning FLUX with DeStyle-100K using lightweight LoRA already achieves comparable results to closed-source models like GPT-4o, further validating the effectiveness of this data construction strategy**.
>
> > Response to W2.
>
> Thank you for the insightful question.
> * In image-driven style transfer [1] [2] [3] [4], **it is a common and widely adopted setting that the stylized output preserves the structural layout of the original content image. Our framework follows this principle**.
> * **Our goal is to build a general-purpose model that supports a wide range of diverse style categories**. For **highly abstract or structure-free styles**, we believe they **may benefit more from specialist models**. Since **our data generation pipeline is modular and flexible, it can be readily extended to support such styles, which we consider a promising direction for future work**.
>
> [1] Image Style Transfer Using Convolutional Neural Networks, CVPR 2016
>
> [2] Domain Enhanced Arbitrary Image Style Transfer via Contrastive Learning, SIGGRAPH 2022
>
> [3] Style Injection in Diffusion: A Training-free Approach for Adapting Large-scale Diffusion Models for Style Transfer, CVPR 2024
>
> [4] OmniStyle: Filtering High Quality Style Transfer Data at Scale, CVPR 2025
>
> > Response to W3.
>
> Thank you for raising this concern.
> * In our construction of DeStyle-100K triplets, **we do not perform global style similarity matching across the entire dataset**. Instead, for each style image, **we select the reference image from within the same style category**, based on the CSD score. This **category-aware matching strategy helps ensure high-quality style matching between the reference and style images**.
> * As shown in Figure 2, the reference image and style image in our triplets exhibit strong stylistic alignment, confirming that our **category-aware matching strategy maintains consistency between reference and target styles**.
>
> > Response to W4.
>
> Thank you for the question.
> * We would like to clarify that **USO is a concurrent ICLR submission, and both works were developed independently around the same time**.
> * Unfortunately, **the destylization model used in USO is not publicly available**, so we are unable to conduct a direct comparison on destylization performance. Therefore, we do compare our method with USO on the downstream stylization task in **Table 3 and Table 5**. **Our method achieves better results across multiple automatic metrics and receives higher user preference scores, which indirectly reflects the effectiveness of our destylization approach**. Moreover, as shown in the **bottom half of Figure 1 and in the qualitative comparisons in Figure 7**, **USO often produces under-stylized results, particularly struggling to handle complex style domains such as papercraft**. In contrast, **our method yields more faithful and expressive stylizations across diverse style types**.
> * To further demonstrate the performance of DestyleNet, we provide **qualitative examples of destylized outputs in Figure 2 of the main paper and Figure 8 in the appendix**. These results show that our method produces **high-quality, pixel-aligned natural content images** from diverse style inputs, supporting the utility of our destylization pipeline.

---

> ### Author Response · Authors · 2025-11-19
>
> > Response to Q1.
>
> Thank you for the suggestion. We have included additional comparison results in the appendix; please refer to Figure 9.
>
> > Response to Q2.
>
> Thank you for the helpful suggestion.
>
> AlignedGen is an excellent and high-quality work, and **we will make sure to cite it in the final version**. However, **AlignedGen focuses on image generation with reference style, rather than image-driven style transfer**. Due to the difference in task formulation and objectives,  **we are unable to make a direct comparison at this stage**.

---

> > ### Comment · Reviewer_NvXS · 2025-11-24
> >
> > Thank you for your rebuttal. While most of my concerns have been addressed, I find the replies to W2 and W4 unconvincing and have therefore decided to maintain my original rating.

---

> > > ### Author Response · Authors · 2025-11-24
> > >
> > > Thank you for your continued engagement and for taking the time to review our rebuttal in detail. We sincerely appreciate your thoughtful comments and are encouraged to know that **most of your concerns have been addressed**.
> > > In the following, we would like to further clarify the remaining questions regarding.
> > >
> > > >  About W2
> > >
> > > * We would like to sincerely clarify that **abstract styles are currently difficult to handle using existing style transfer models**, primarily due to their complex and unconventional visual structures.
> > > * However, **our destylization-based approach is not in conflict with abstract styles**—in fact, **it is inherently compatible**. In our pipeline, we leverage existing stylization models to construct paired data for training the destylization model DestyleNet. Once a stylization model becomes capable of transferring abstract styles, we can use it to synthesize corresponding destylization training pairs and extend our method to abstract domains.
> > > * In fact, **our current model already supports abstract styles to a certain extent**. For example, **the upper-right result in Figure 2 of the main paper** and **the first row of Figure 10 in the appendix** demonstrate our method’s ability to destylize and stylize inputs with abstract visual styles. Notably, the results in the first row of Figure 10 show that our method achieves higher-quality abstract-style transfer compared to USO, GPT-4o, and Qwen-Image-Edit.
> > >
> > > > About W4
> > >
> > > Thank you again for your thoughtful follow-up.
> > > Regarding W4, we would be grateful if you could kindly clarify which specific aspect remains a concern. We are more than willing to provide further clarification or incorporate additional evidence to resolve any remaining questions.
> > >
> > > We truly appreciate your thoughtful engagement and look forward to any additional suggestions that could help further improve our work.

---

> > > ### Author Response · Authors · 2025-11-27
> > >
> > > Dear Reviewer NvXS,
> > >
> > > We truly appreciate your comments and the time you have spent reviewing our work. We hope our previous response has addressed your concerns regarding W2. **Regarding W4, we would be sincerely grateful if you could kindly clarify which specific aspects remain a concern**. We would be glad to provide further explanations or supporting evidence. Your insights are very important to us, and we warmly welcome any additional suggestions that could help improve the paper.

---

> > > > ### Comment · Reviewer_NvXS · 2025-11-27
> > > >
> > > > For W2, I think it is the limitation that the performance of your model is depends on current SOTA style transfer methods, which also metioned by other reviewers.
> > > > And for W4, could you provide a more comprehensive, fundamental comparison or analysis with USO destylization—for example, discussing advantages in terms of model architecture, dataset and related aspects? Because it is the most relevant work to your method.

---

> > > > > ### Author Response · Authors · 2025-11-28
> > > > >
> > > > > We are very grateful for your kind follow-up and your willingness to engage in further discussion.
> > > > >
> > > > >
> > > > > > **About W4: fundamental comparison or analysis with USO destylization**
> > > > >
> > > > > Thank you for your thoughtful question. **We regard USO as an excellent concurrent work that makes meaningful contributions to the field**.
> > > > >
> > > > > * Unfortunately, the **USO paper does not provide sufficient details regarding the architecture, training data, or optimization objectives of their destylization module**. We note that **this lack of information has also been mentioned by ICLR reviewers of the USO paper**. As a result, a direct and rigorous comparison between our DeStyleNet and the destylization model used in USO is unfortunately infeasible.
> > > > >
> > > > > * Based on the limited destylized results presented in Figure 2 of the USO paper (only three examples), we observe several issues:
> > > > >    - **Lack of pixel-aligned structure preservation**: For example, in the *Bamboo Painting* case, both the foreground insect and background leaves shift significantly in position and scale after destylization.
> > > > >    - **Loss of essential content elements**: In the *Vegetable Market* example, objects such as vegetables in the bottom-left corner are entirely missing in the destylized result, and background characters are altered or scaled inconsistently.
> > > > >    - We emphasize that **maintaining structural alignment and preserving original content** are critical prerequisites for style transfer tasks.
> > > > >
> > > > > * By contrast, **our method provides high-quality, pixel-aligned destylized results across 19 diverse cases (see Figure 2 and Figure 8), spanning a wide range of artistic styles (e.g., oil painting, abstract, flat design, pixel art, origami, 3D rendering) and semantic categories (e.g., human, animal, plant, scene, architecture)**. Importantly, **all destylized outputs retain the content elements and structure of the original style images**—providing a reliable foundation for training.
> > > > > * Furthermore, we have added **new quantitative evaluations** of DeStyleNet’s destylization quality using multiple automated metrics (Appendix, Lines 831–858), which further support the effectiveness of our approach.
> > > > > * Finally, **our stylization results—both qualitative and quantitative—consistently outperform USO**. This indirectly supports the quality of our destylization pipeline as well.
> > > > >
> > > > > Taken together, we believe the above evidence demonstrates that our destylization method is effective, scalable, and well-suited for constructing high-quality training data for image-driven style transfer.
> > > > >
> > > > > ---
> > > > >
> > > > > > **About W2: Limited by SOTA Style Transfer Methods**
> > > > >
> > > > > We sincerely appreciate your insightful comments.
> > > > >
> > > > > * We would like to **respectfully** clarify that **our method is not limited by current SOTA style transfer models**. As direct evidence, in **Table 2** and **Table 4**, our method consistently outperforms several strong style transfer baselines—**including CSGO, StyleID, AD, and STROTSS—even though these methods were used to construct the initial training data for destylization**. This demonstrates that our pipeline not only avoids inheriting their biases but also surpasses their capabilities through improved supervision quality.
> > > > >
> > > > > * Moreover, we believe that our destylization-based approach **offers a scalable and practical solution** for handling more complex or previously unsupported styles. For example, by leveraging powerful image editing models like **Gemini-2.5-Flash-Image** to create a small set of high-quality triplets, we can use these data to fine-tune our DeStyleNet via lightweight PEFT methods (e.g., LoRA), which then enables the generation of more high-quality data for complex styles.
> > > > >
> > > > > **We sincerely hope this clarification addresses your concerns. If any questions remain unresolved, we are more than willing to provide further evidence or make any necessary improvements.**

---

### Official Review · Reviewer_jeZr · 2025-10-22

**Soundness:** 2
**Presentation:** 2
**Contribution:** 3
**Rating:** 4
**Confidence:** 2

**Summary:**

This paper proposes a data-centric method for image stylization to address its ill-posed nature, where the ground-truth stylized image is often absent for a specific content reference image. While previous work often relies on synthetic stylized images as learning targets, the authors propose to synthesize the input content image from a human-created artwork to construct paired data with authentic training objectives. This is achieved by training an image de-stylization model, named DestyleNet, that removes stylistic elements from artworks to recover natural images. Leveraging this model, the authors curate the DeStyle-100K dataset consisting of 100K <destyle image, style reference, target image> paired data. To ensure high data quality, they also propose an automatic pipeline called DeStyleCoT-Filter, using MLLM to filter out data pair with either content or style mismatch. Finally, they introduce an image stylization benchmark BCS-Bench with balanced style and content diversity.

**Strengths:**

- The motivation behind the proposed method is clear. Due to the absence of paired data, current model-centric image stylization methods rely on synthetic data for end-to-end training, where the stylized target images are largely generated and curated through sophisticated data processing pipelines. The authors propose to, instead of synthesizing the learning target, synthesize the input content image. Such approach ensures high-quality in the learned data distribution, while the artifacts in the synthetic content images can effectively augment input data to improve the robustness of the model. The entire process is reasonable.
- The proposed image de-stylization method is interesting. For training the de-stylization model DestylNet, the authors construct an inversed problem and leverage existing image stylization methods with the condition and target switched. Such a design is intuitive and effective.
- The proposed data curation pipeline is well designed. Both the dataset and benchmark will benefit future work in this field.

**Weaknesses:**

- In the construction of the DeStyle-100K dataset, the authors collect 10K artworks from the internet. To improve data diversity the authors additionally generate 150K stylized images using FLUX which is extremely unproportionate to the curated artwork data. Even after post processing by DestyleCoT-Filter, the total amount of 100K images means the synthetic data consists of **at least 90\%** of the training dataset. This imbalance stands in stark contrast to the purpose of DeStyle-100K and diminishes the effectiveness of the proposed method from my perspective.
- In line 275-276 Section 3.4, the authors use MLLM to analyze the style discrepancy in the synthesized data pairs, where the style of an image is decomposed into attributes like color palette, texture, lighting and rendering effects. However, there lacks a detailed explanation for this step. How was the decomposition operated on the image? Is it also processed by MLLM like GPT?
- In quantitative comparisons, the authors use the textual description of the style reference as a proxy to condition text-guided models like Qwen-Image-Edit. Such setting is potentially adversarial for these methods and thus risks of unfair comparison. Combined with the imbalanced training data, the evidence for supporting the effectiveness of the proposed method is quite limited.
- Limitation is not discussed. From my point of view an obvious limitation will be the absence of text prompts in the synthesized data, and in the proposed DeStyle2Style model.
- There are several typos, especially in the citation style that seems unaligned with ICLR author guidelines. For instance: line 205, 211, 240, 247.

**Questions:**

- To improve the data diversity of the DeStyle-100K dataset. Wouldn’t it be simpler to employ the 60K triplets training data of DestyleNet, replacing the stylized target with the original style reference image and vice versa? In this way we could have both: 1) authentic learning target, since the target is now reference style image; 2) diverse input content images, including both the original 10K natural image curated in DeStyle-100K dataset, as well as digital art in the 60K triplets.
- For artworks like abstract art, Piet Mondrian for example. The content of these images can not be clearly defined, yet in other stylization pipelines they are eligible for serving as reference. I wonder how the proposed method generalizes to such data.

---

> ### Author Response · Authors · 2025-11-19
>
> Thank you for your thoughtful review and constructive feedback. We sincerely appreciate your comments **highlighting the clear motivation, the reasonable architectural design, the interesting and effective de-stylization method, and the well-designed data curation pipeline that may benefit future work in this field**. Below, we address each of your concerns individually.
>
> > Imbalanced Real–Synthetic Data Ratio.
>
> Thank you for the comment.
> * Although many of our style images are generated using a text-to-image model (FLUX-Dev), their **visual quality is already comparable to real artworks**. Moreover, these **AI-generated style images** have been increasingly **recognized and adopted by artists in real-world creative workflows**.
> * Crucially, **we do not use synthetic stylized images as supervision targets like OmniStyle and CSGO**, which often contain **artifacts and style leakage**. Our method is explicitly designed to avoid such artifacts by using **high-quality style images**, whether real or T2I-generated, **as the sole supervision signal**. This ensures reliable training and strong performance.
>
> > Clarification on Style Attribute Decomposition.
>
> Thank you for your question.  Yes, the decomposition of style attributes is performed by an MLLM (e.g., GPT-4o) given the input style image, guided by carefully designed prompts to extract attribute-level descriptions such as color palette, texture, lighting, and rendering effects.  Due to space limitations, we have included a detailed example of this process in the appendix, please refer to Figure 15.
>
> > About Using Text Descriptions for Comparison with Image Editing Models.
>
> Thank you for the comment.
> * **Using text descriptions as a proxy for style input is a deliberate design choice made for fairness and compatibility**. Models such as **Qwen-Image-Edit, Bagel, and FLUX-Kontext do not support multi-image conditioning**. Forcing both style and content images as **concatenated inputs significantly degrades their performance**, which would lead to an **unfair evaluation**.
> * In addition, **we include comparisons with models that support multi-image input, including the closed-source GPT-4o, as well as recent state-of-the-art methods such as USO, OmniStyle, and StyleID**. Our method outperforms OmniStyle, StyleID, and USO, and achieves performance comparable to GPT-4o, as shown in Tables 2-5.
>
> > Limitation: Absence of Text Prompts.
>
> Thank you for your insightful comment.
> * Due to space limitations in the main paper, we have provided a discussion of this limitation in Appendix A.1: Limitations and Future Work, and we kindly refer the reviewer to that section for details.
> * Our method **follows the standard setting of reference-based style transfer, where a style image is used as condition input rather than a text prompt**. This design is consistent with a wide range of existing works [1] [2] [3] [4], all of which rely on style reference image to guide stylization.
> * In addition, the DeStyle-100K dataset provides text-related annotations, including:
>   - The **artist name and art movement** (see Appendix Table 8 and Table 9),
>   - The **style category and its corresponding style description** for each style image.
>
> * Moreover, since the dataset provides well-aligned triplets (style, content, and stylized result), it can also be extended to a text-supervised setting by using MLLMs to efficiently generate instruction-style prompts.
>
> [1] Image Style Transfer Using Convolutional Neural Networks, CVPR 2016
>
> [2] Domain Enhanced Arbitrary Image Style Transfer via Contrastive Learning, SIGGRAPH 2022
>
> [3] Style Injection in Diffusion: A Training-free Approach for Adapting Large-scale Diffusion Models for Style Transfer, CVPR 2024
>
> [4] OmniStyle: Filtering High Quality Style Transfer Data at Scale, CVPR 2025
>
> > Formatting Issue: Citation Style and Typos.
>
> Thank you for your careful reading and helpful suggestions. We have revised the main paper to correct the citation formatting and fix the typos you pointed out (see the blue highlights around lines 205, 212, 242, and 249). We have also thoroughly proofread the entire manuscript to ensure consistency and compliance with the ICLR author guidelines.
>
>
> > About Using DestyleNet Triplets as Stylization Supervision.
>
> Thank you for the thoughtful suggestion.
> * However, the 60K triplets used to train DestyleNet cannot provide authentic supervision for stylization training. The **stylized images in these triplets are synthesized by existing style transfer models, which often contain visual artifacts and style drift, making them unsuitable as training targets**.
> * Moreover, **although the style reference images in those triplets are real, they are not paired with corresponding content images, and therefore cannot serve directly as ground-truth targets**. This limitation underscores the necessity of **our destylization-based construction pipeline, which ensures reliable content-style alignment and authentic supervision**.

---

> ### Author Response · Authors · 2025-11-19
>
> > About Handling Abstract Artworks.
>
> Thank you for the thoughtful question.
> * We agree that artworks such as those by Piet Mondrian or Picasso represent abstract or conceptual styles that do not align with real-world semantics. **These artworks are often driven by artistic imagination rather than depictive realism, making it difficult to define or extract their underlying "content."**
> * **Our method is fundamentally data-driven**. If existing stylization methods were able to handle such abstract styles reliably, they could in principle be used within our data generation pipeline to train DestyleNet. However, we observe that most current methods struggle to process abstract styles.
> * **Our goal is to build a general-purpose model that supports a diverse range of styles**. For highly **abstract or conceptual styles,** we believe it is **more appropriate to design specialist models tailored to their unique characteristics**, which we consider an important direction for future work.

---

> > ### Comment · Reviewer_jeZr · 2025-11-26
> > **Official Comments by Reviewer jeZr**
> >
> > I appreciate the authors' efforts for clarifying questions and weaknesses. Most of my concerns are resolver. However, the authors' comment on **weakness 1 "Imbalanced Real–Synthetic Data Ratio."** does not resolve the contradiction of their claim, which remains as the major weakness.
> >
> > Although the SOTA models like FLUX already produce high-quality images, there is a fundamental gap between their generations and real-world images, especially **in the domain of artistic creation**. However, in lines 97-99, the authors claim their major contribution as
> >
> > > DeStyle2Style reframe artistic style transfer as a data generation problem, addressing the fundamental limitation of absent authentic supervision.
> >
> > However, as the original weakness 1 stated, **90~94% of DeStyle-100K are generated by FLUX.** Such a substantial amount of generated images significantly diminishes the claim that the advantage of the proposed lies in using **authentic artworks** as training target. In conclusion, **the proposed method is sound if the contribution and narrative flow can be properly stated.**

---

> > > ### Public Comment · ~Anay_Majee3 · 2025-11-26
> > >
> > > -  I agree with your point. I think this is due to the inherent limitations of the dataset construction method and the model itself. Therefore, it overly relies on the training samples and fails to achieve true style removal.

---

> > > > ### Author Response · Authors · 2025-11-27
> > > >
> > > > Thank you for joining the discussion.
> > > >
> > > > We would like to clarify several points:
> > > >
> > > > 1. **Our goal is not to achieve “true style removal”**. Instead, **we aim to recover natural, style-reduced counterparts that retain structural and semantic fidelity**, as clearly stated throughout our paper (e.g., lines 34, 38, 81, 183). As demonstrated in Figure 2 of the main paper and Figure 8 of the appendix, our method performs well **not only on simple styles but also on more complex and structured ones such as 3D renderings, origami, and flat design, all while preserving pixel-level alignment with the original content**.
> > > > 2. **Our data construction pipeline is not inherently limited**; rather, **it is scalable and flexible**. DestyleNet can be applied to real artworks such as **Monet and Van Gogh** (see Figure 2 and the first row of Figure 8), and even works for **abstract artistic styles** (e.g., the top-right sample in Figure 2), showing its generalization beyond synthetic inputs.
> > > > 3. We use more FLUX-generated images in our dataset because **collecting a sufficiently large-scale and stylistically diverse set of real artworks is impractical in resource-constrained scenarios**. In contrast, FLUX-based T2I generation offers an efficient and controllable alternative without compromising visual quality.

---

> > > > > ### Comment · Reviewer_jeZr · 2025-11-27
> > > > > **Official Comments by Reviewer jeZr**
> > > > >
> > > > > Thanks for the authors' clarification and the acknowledgment of the authentic artistic creation. I believe there is still gap between the outputs of SOTA T2I models and genuine artworks. Considering this paper's exploration towards destylization and the corresponding dataset, I've decided to increase my original rating if the main contributions and the relevance part in the manuscript are properly revised.

---

> > > > > > ### Author Response · Authors · 2025-11-27
> > > > > >
> > > > > > We sincerely appreciate your insightful comments and constructive feedback. Following your suggestions, **we have carefully revised the relevant parts of the manuscript** to better address your concerns. Below are the key changes made:
> > > > > >
> > > > > > 1. We have **added a clear definition of authentic supervision in the introduction** to ensure it is properly explained and fully contextualized. Please refer to **lines 72–79** in the revised manuscript.
> > > > > > 2. The descriptions in the **contributions section have been refined** for greater clarity and accuracy, **ensuring that our main contributions are properly stated**. Please refer to **lines 99–106** in the revised manuscript.
> > > > > > 3. In the **Method section**, we have **further elaborated on the definition of authentic supervision and clarified the sources of data used**. Please refer to **lines 198–202** in the revised manuscript.
> > > > > >
> > > > > > We truly thank you for your valuable feedback. Your suggestions have helped us better articulate the core ideas of our work and strengthened its overall presentation. **If there are any remaining concerns or additional questions, we are more than willing to address them in full.**

---

> > > > > > > ### Author Response · Authors · 2025-11-28
> > > > > > >
> > > > > > > Dear Reviewer,
> > > > > > >
> > > > > > > Thank you again for your valuable time and feedback. We truly appreciate your earlier comments and suggestions. We would like to kindly check whether our previous response has adequately addressed your concerns.
> > > > > > >
> > > > > > > If there are any remaining questions or issues, we are more than willing to provide further clarification or evidence. We sincerely look forward to your continued feedback and to further constructive discussion.

---

> > > ### Author Response · Authors · 2025-11-27
> > >
> > > Thank you again for your constructive insights. We truly appreciate your thoughtful reminder, and **will revise the paper to ensure that our definition of authentic supervision and the overall narrative of contributions are stated clearly, consistently, and with sufficient grounding**.
> > >
> > > We would like to take this opportunity to further clarify our **definition of authentic supervision** in the context of our pipeline.
> > >
> > > * In previous works such as OmniStyle and CSGO, the **supervision targets** used for training style transfer models are typically **pseudo-stylized images—i.e., content images modified by a style transfer model**. These pseudo targets often suffer from **entangled style-content signals** and **visual artifacts**, making them **unreliable** for high-quality supervision.
> > > * In contrast, our notion of **authentic supervision refers to using unmodified style images as the learning target**, including both real artworks and high-quality style images synthesized by FLUX-Dev, which is trained on real-world distributions. These images are directly generated from text prompts, **not modified from existing content images**, and they exhibit **high visual clarity, strong stylistic expression, and minimal artifacts**. By relying on such style exemplars, our pipeline avoids the limitations of artifact-prone pseudo supervision and provides high-quality supervision signals for training generalizable style transfer models.
> > >
> > > To further clarify, we now elaborate on **why we choose to synthesize style exemplars using FLUX T2I**.
> > >
> > > * Many style categories—such as origami, voxel art, and flat illustration—are difficult to collect in sufficient quality and quantity from real artworks. In these cases, FLUX-generated style images offer a practical and scalable solution. In practice, we observe that FLUX outputs can achieve comparable visual quality to those generated by proprietary models such as GPT-4o and Gemini 2.5 Flash-Image, making it a strong candidate for authentic and efficient supervision, especially in resource-limited settings.
> > > * **Our pipeline is scalable and flexible**, as long as a sufficient number of diverse real artworks can be collected, our destylization-based method is fully capable of utilizing them to train robust stylization models.
> > >
> > > Thank you again for your constructive insights. We are glad to provide further clarifications or make additional revisions based on any remaining concerns or suggestions.

---

### Official Review · Reviewer_pxZ2 · 2025-10-31

**Soundness:** 3
**Presentation:** 3
**Contribution:** 3
**Rating:** 4
**Confidence:** 5

**Summary:**

DeStyle2Style introduces a novel data-centric approach by reversing style transfer through "destylization" to create a large-scale, aligned dataset called DeStyle-100K.

It develops two key models, DestyleNet for reconstructing content images and DestyleCoT-Filter for automatic quality control, alongside a new balanced benchmark, BCS-Bench, for evaluation.

This paradigm of generating supervised data via destylization effectively overcomes the fundamental challenge of lacking ground-truth data in artistic style transfer.

**Strengths:**

1.The method is very clear and the description is reasonable.
2.The dataset pipeline is quite clear.
3.The results achieved are quite good.

**Weaknesses:**

The authors did not clearly explain why the reverse-constructed data pipeline achieves better performance—whether it is due to the advantages of the data itself or the enhancement brought by the reverse data construction. The ablation experiments here are not clear. Besides, the authors did not provide SDXL-based experimental evidence to verify the reliability of the data pipeline. The number of parameters here may be a more critical factor affecting performance. Additional explanations are needed.
There are some printing errors in Table 2; please note this. The authors do not seem to have considered more improvements to the framework. Are VAE features important?

**Questions:**

see Weaknesses

---

> ### Author Response · Authors · 2025-11-19
>
> Thank you for your thoughtful review and constructive feedback. We sincerely appreciate your comments **highlighting the clarity of our method, the reasonableness of the description, and the good results achieved**. Below, we address each of your concerns individually.
>
> > Why the reverse-constructed data pipeline achieves better performance？
>
> Thank you for the question.
> * Our reverse pipeline achieves superior stylization performance primarily due to the use of **high-quality, authentic supervision signals**. We transform real artworks into natural content images via destylization, **enabling the original artwork itself to serve as the sole supervision target for stylization training**.
> * In contrast, **forward pipelines (OmniStyle, CSGO)** rely on **synthetic pseudo-targets** generated by imperfect style transfer models, which inevitably **introduce supervision noise such as style leakage and visual artifacts**.
> * As shown in Table 2 and Table 4, **our method consistently outperforms these forward-based approaches across multiple metrics**, validating the effectiveness of our reverse-constructed data pipeline.
>
> > SDXL-based experimental evidence.
>
> Thank you for the helpful suggestion.
>
> * While our experiments are based on the FLUX-Dev architecture, we apply **parameter-efficient LoRA fine-tuning**, **rather than full fine-tuning**. **OmniStyle** also adopts the same FLUX-Dev architecture, but uses **full fine-tuning**, with **approximately 20× more trainable parameters**. Despite this, **our method achieves better overall performance, which highlights the effectiveness of our data pipeline—suggesting that training on synthetic pseudo-targets, as in OmniStyle, may limit performance regardless of parameter scale**.
> * We also provide both **quantitative results of SD3-medium (2B), a smaller model than SDXL (2.6B), fine-tuned on DeStyle-100K**. The quantitative comparison is provided in the table below. Despite SD3-Medium having a parameter size six times smaller than Flux-Dev, its performance does not degrade significantly. In fact, it even outperforms Flux-Dev on certain metrics, such as Style Loss and Aesthetic. This demonstrates that the model effectively leverages the data pipeline, maintaining strong performance even with reduced parameter sizes.
> * Furthermore, qualitative results and deeper analysis can be found in Figure 12, Table 6, and Section A.3.4 (Appendix). These results strongly highlight the generalizability and architecture-agnostic reliability of our data pipeline.
>
> | Backbone     | Parameters | DINO Score   | CLIP Score   | CSD Score    | Style Loss | Qwen-Content  | Qwen-Style    | Qwen-Aesthetic |
> |--------------|------------|--------------|--------------|--------------|------------|----------|----------|-----------|
> | Flux-Dev     | 12B        | **0.8203**   | **0.2702**   | **0.5606**   | 0.1170     | 8.1385   | **7.5763** | 8.7326    |
> | SD3-Medium   | 2B         | 0.7473       | 0.2356       | 0.5341       | **0.0518** | **8.4413** | 7.4789   | **9.2032** |
>
>
> > About some printing errors.
>
> Thank you for pointing this out. To improve clarity, we have separated the user-study quantitative results from Table 2 and Table 3. The corresponding results are now presented as two standalone tables (Table 4 and Table 5), making the comparison more explicit and easier to follow.
>
> > About framework design choices.
>
> Thank you for the question.
> * To fairly demonstrate the value of the DeStyle-100K dataset and the effectiveness of our destylization-based pipeline, we adopt the FLUX-Dev architecture **without introducing additional modules and keep the VAE frozen during training**.
> * **This minimal setup isolates the contribution of the data itself and already yields strong stylization performance, competitive with GPT-4o, Qwen-Image-Edit, and FLUX-Kontext (Table 3 and Table 5)**. While fine-tuning the VAE may further improve performance, it requires significant computational resources and is left for future work.

---

> > ### Author Response · Authors · 2025-11-27
> >
> > Dear Reviewer pxZ2,
> >
> > Thank you again for your earlier insightful comments. We sincerely hope to address any remaining concerns you may have. If there are any questions we haven’t clarified or further suggestions you'd like to share, we would greatly appreciate the opportunity to respond and improve the work accordingly.

---

> > > ### Comment · Reviewer_pxZ2 · 2025-11-27
> > >
> > > "Sorry for the delay. I acknowledge the authors' rebuttal efforts, but I still share the concerns raised by Reviewer jeZr regarding your claims. Specifically, despite using extensive synthetic images, the difference from OmniStyle and CSGO does not seem significant.
> > >
> > > I have questions regarding Figure 4(b): Are the VAE features critical, and how is the feature fusion performed? Is your structure identical to OmniStyle? The data advantage also feels marginal. I await your further response."

---

> ### Author Response · Authors · 2025-11-27
>
> Thank you very much for your thoughtful follow-up and for taking the time to engage further with our work. Below, we provide point-by-point clarifications to address your concerns.
>
> > About the difference from OmniStyle and CSGO
>
> Thank you for raising this important point. Compared to OmniStyle and CSGO, we would like to emphasize that our method introduces a **fundamentally different data construction paradigm**.
>
> * **OmniStyle and CSGO** construct training triplets by **applying style transfer models to edit natural content images, generating stylized results as learning targets.** However, these stylized targets are inherently **pseudo, often suffering from artifact noise and entangled style-content signals, which compromise the reliability of supervision**.
>
> * In contrast, **our pipeline adopts a reversed formulation**: **we start from style images and apply destylization to obtain style-reduced natural content**. This enables us to **use the unmodified style image**—whether it is a real artwork or a high-quality style image generated by FLUX-T2I—**as the sole supervision signal**. The destylized output is used as the content input and does not affect the supervision signal itself.
>
> This design ensures that the style supervision is both authentic and clean, avoiding the artifacts and leakage issues common in stylization-based pipelines such as OmniStyle and CSGO. We believe this core distinction provides more reliable data support for training controllable and high-fidelity stylization models.
>
> > About the data advantage
>
> Thank you for your valuable comment. We would like to **respectfully** clarify that **the advantages of our data pipeline are indeed significant**, and we provide the following evidence to support this:
>
> * **Strong performance with smaller-scale data and lightweight training.** Compared to OmniStyle, which utilizes a massive dataset of over 100M stylized pairs and performs full-parameter fine-tuning, our method only uses 100K training triplets and applies LoRA-based fine-tuning with approximately 1/20th the trainable parameters. Despite this, as shown in Table 2 and Table 4, our method consistently outperforms OmniStyle in both quantitative metrics and human evaluation. This demonstrates the superior quality and supervision signal of our data.
>
> * **Outperforming strong open-source models and matching proprietary models**. Although our stylization model is trained solely on our 100K dataset, it achieves higher scores than state-of-the-art open-source image editing models such as Qwen-Image-Edit and Flux-Kontext, and in many cases achieves comparable results to GPT-4o (see Table 3 and Table 5). These results provide further evidence of the effectiveness and generalizability of our data.
>
> * **Effective performance across different model scales.** We additionally fine-tuned SD3-Medium (2B) using our DeStyle-100K dataset. The results—presented in Figure 12 and Table 7 in the appendix—demonstrate that our data pipeline continues to deliver good stylization performance even on smaller backbone models. This further supports the advantage and generalizability of our dataset, which does not rely on specific model architectures or parameter scales.
>
> In summary, despite using fewer data and lighter training, **our method consistently outperforms or matches much larger and stronger baselines**. This clearly illustrates the effectiveness and advantage of our destylization-based data construction pipeline.
>
>
> > Are the VAE features critical, and how is the feature fusion performed?
>
> Yes, **the VAE features play a critical role in our pipeline**. As illustrated in Figure 4b, the content and style images are first encoded into latent representations using the VAE encoder, resulting in content latents and style latents, respectively. The content latents are then perturbed with noise to obtain noisy latents. These noisy latents are **spatially concatenated** with the style latents to **form a unified latent sequence**, which is subsequently fed into the DiT backbone for generation.
>
>
> > Is your structure identical to OmniStyle?
>
> Our method adopts the **same FLUX-Dev backbone** as OmniStyle. **We intentionally chose to use this simple and unmodified backbone to better isolate and highlight the effectiveness of our proposed data pipeline. This design choice allows us to validate the impact of data quality on stylization performance without introducing additional architectural factors.**
>
> **We deeply appreciate and respect your careful review and insightful feedback. If any concerns remain, we would be more than willing to provide further clarification or additional evidence to support a constructive discussion.**

---

> > ### Author Response · Authors · 2025-11-28
> >
> > Dear Reviewer,
> >
> > Thank you again for your valuable time and feedback. We truly appreciate your earlier comments and suggestions. We would like to kindly check whether our previous response has adequately addressed your concerns.
> >
> > If there are any remaining questions or issues, we are more than willing to provide further clarification or evidence. We sincerely look forward to your continued feedback and to further constructive discussion.

---

> > ### Comment · Reviewer_pxZ2 · 2025-11-28
> >
> > Regarding de-stylization, how do you ensure the style is removed cleanly without introducing new style leakage or altering the content? What guarantees this?
> >
> > I understand the authors want to emphasize the data advantage, but relying solely on written descriptions is hardly convincing.

---

> > > ### Author Response · Authors · 2025-11-28
> > >
> > > Thank you for your valuable comment.
> > >
> > > >  Regarding de-stylization, how do you ensure the style is removed cleanly without introducing new style leakage or altering the content? What guarantees this?
> > >
> > >
> > > First, we demonstrate that our destylization model, **DeStyleNet, inherently produces high-quality, structure-aligned, and style-reduced images**. To validate this, we conduct quantitative evaluations across five randomized test sets (see Appendix, Lines 831–858). Specifically, we conducted **5 independent trials, each randomly sampling 1,000 destylized images, and evaluated them using 4 comprehensive metrics**:
> > >
> > > * **ID Score**: Measures identity consistency between the destylized image and its original content image.
> > > * **Style Removal Score**: Quantifies the extent to which style has been effectively removed.
> > > * **Image Quality**: Assesses the image quality of the destylized output.
> > > * **Image Aesthetic**: Reflects the overall aesthetic appeal of the destylized result.
> > >
> > > All metrics were computed using QwenVL-Max, prompted appropriately for each evaluation dimension. The scoring range for all metrics is standardized to **0–5, where higher values indicate better performance**.
> > >
> > > | Test Set | ID Score | Style Removal Score | Image Quality | Image Aesthetic |
> > > |----------|----------|---------------------|---------------|------------------|
> > > | Set 1    | 4.9467   | 4.0046              | 4.6123        | 4.5120           |
> > > | Set 2    | 4.9397   | 3.9755              | 4.6123        | 4.5267           |
> > > | Set 3    | 4.9190   | 3.9770              | 4.6229        | 4.5218           |
> > > | Set 4    | 4.9314   | 3.9545              | 4.6202        | 4.5100           |
> > > | Set 5    | 4.9358   | 3.9219              | 4.6193        | 4.5210           |
> > > | **Mean** | **4.9345** | **3.9667**         | **4.6174**    | **4.5183**       |
> > >
> > > Across 5 independent trials (each with 1,000 randomly sampled destylized images), our method achieved a **mean ID Score of 4.9345, a Style Removal Score of 3.9667, an Image Quality score of 4.6174, and an Image Aesthetic score of 4.5183**. These results demonstrate that **our method inherently achieves effective style removal while faithfully preserving identity and structure**.
> > >
> > > ---
> > >
> > > Second, we **further ensure output quality** via a well-designed filtering module called **DeStyleCoT-Filter** (see Main Text, Lines 291–295), which uses GPT-4o to assign:
> > >    - **Content Preservation Score**, evaluating how well the destylized image aligns structurally with the original stylized artwork;
> > >    - **Style Discrepancy Score**, measuring how much stylistic information has been removed.
> > >
> > > Only samples that achieve **high scores in both dimensions** are retained for dataset construction. This dual-criteria filtering further ensures that the final training triplets are clean and reliable.
> > >
> > > ---
> > >
> > > We sincerely hope the explanation above addresses your concerns. **If further clarification or evidence is needed, we would be delighted to include more data cases from our DeStyle-100K dataset in the Appendix for your thoughtful consideration**.

---

> > ### Public Comment · ~Anay_Majee3 · 2025-11-28
> >
> > > The majority of the data you constructed is generated by AI, and the results of its generation may not accurately represent real-world images. In other words, the generated ground truth (GT) might be a representation of another stylized data. Therefore, based on this point, it may not be feasible to achieve what you refer to as 'de-stylization'.

---

### Public Comment · ~Anay_Majee3 · 2025-11-26

Hi, author. I'm also interested in style transfer. As Reviewer jeZr pointed out, this is indeed the primary limitation of the current work. We further raise two additional concerns:
>1. The outputs after de-stylization still exhibit significant discrepancies from natural images, suggesting that photorealism has not been fully achieved.

> 2. The performance on out-of-domain (OOD) data remains unclear. specifically, it is uncertain whether the model retains its effectiveness when applied to unseen domains.

In conclusion, due to limitations in both the dataset construction methodology and the model architecture, the method may not have truly accomplished effective de-stylization.

---

### Author Response · Authors · 2025-12-01
**Rebuttal Summary for the AC**

Dear Area Chair,

Thank you very much for taking the time to coordinate the review process of our submission. To assist you in your assessment, we would like to briefly summarize the main points of the rebuttal process and our efforts to address reviewer concerns.

### **Addressing Reviewer Concerns:**

First, we would like to sincerely note that **before the information leak occurred**, *Reviewer jeZr* had already expressed a willingness to raise their original score (which was a 4). In response, we carefully implemented the requested changes to the **contribution statement, introduction, and method sections**.

Throughout the rebuttal phase, we actively and thoroughly responded to each comment with detailed explanations and supporting quantitative results. Below, we summarize our responses to each reviewer:

1. **Reviewer pxZ2 (Score: 4):** We addressed all four concerns raised in the first round and provided detailed responses and quantitative experiments for the additional issues raised during the second and third rounds.

2. **Reviewer jeZr (Score: 4):** We addressed seven key concerns in the first round and revised the paper in accordance with the reviewer’s suggestions. **The reviewer acknowledged these efforts and indicated a willingness to raise their score**.

3. **Reviewer NvXS (Score: 6):** This reviewer confirmed that most of their concerns were resolved in the first round. For the remaining issues, we submitted additional evidence and discussion in our follow-up response.

4. **Reviewer DF8q (Score: 6):** We addressed both of this reviewer’s concerns in our initial response and further supported our claims with quantitative results in the follow-up.

---

### **Our Contributions**

1. **DeStyle2Style Framework:** We propose a novel destylization-based pipeline that reframes artistic style transfer as a data construction problem.  It enables the use of unaltered style images as direct learning targets through de-stylization, providing high-quality supervision signals for the style transfer.

2. **DeStyle-100K Dataset:** We introduce a large-scale dataset consisting of 100K high-quality triplets built through destylization. In contrast to prior datasets with pseudo-stylized targets, DeStyle-100K offers *authentic supervision*, where unaltered style images directly guide the training process.

3. **DestyleCoT-Filter:** To ensure the quality of constructed data, we design a fine-grained CoT-based evaluation framework that jointly enforces *content preservation* and *style discrepancy* to verify the effectiveness of each sample.

4. **BCS-Bench Benchmark:** We build a balanced benchmark for evaluating style transfer methods, comprising 56 style images across 35 artistic styles and 55 content images across 6 semantic categories (human, animal, plant, scene, architecture, object), forming 3,080 diverse content-style pairs for both quantitative and qualitative evaluation.

---

We sincerely hope our destylization-based paradigm will contribute to the community and inspire further research in style transfer.

Thank you again for your time and consideration.

Sincerely,

The Authors

---

### Meta-Review · Area_Chair_CtDv · 2025-12-29

**Summary:**

This paper proposes DeStyle2Style, a framework that reframes style transfer as a data generation problem by using a destylization model (DestyleNet) to reverse-engineer natural content from artworks, thereby creating a large-scale dataset (DeStyle-100K) with authentic supervision signals.
Reviewers appreciated the novel data-centric perspective of using destylization to obtain paired data, the comprehensive construction of the pipeline and benchmark (BCS-Bench), and the clear presentation of the motivation.

The reviewers identified several critical concerns:
1. **Reliance on Synthetic/Pseudo Data**: Reviewers questioned whether DestyleNet, being trained on data synthesized by existing style transfer models, inherently learns the biases and limitations of those base models, preventing true destylization. (DF8q, NvXS, pxZ2)
2. **Dataset Authenticity and Imbalance**: Reviewers noted that over 90% of the style images in DeStyle-100K are generated by FLUX rather than being real artworks, which contradicts the claim of authentic supervision and authentic artistic creation. (jeZr, pxZ2)
3. **Handling Abstract Styles**: Concerns were raised regarding how the method handles abstract art where content is undefined, as destylization typically assumes structure preservation which might not apply to abstract domains. (NvXS, jeZr)
4. **Comparison and Evaluation**: Reviewers requested comparisons with other destylization methods (e.g., USO) and questioned the lack of quantitative metrics specifically evaluating the quality of the destylized outputs. (NvXS, DF8q, pxZ2)

In summary, this paper was reviewed by four experts in the field. The recommendations are 4, 4, 6, 6. The reviewers raised concerns regarding the heavy reliance on FLUX-generated data, the potential for the model to inherit biases from the synthetic training data, and the handling of abstract styles. After rebuttal, while the paper has merit in its fresh perspective and large-scale dataset construction , concerns on the validity of synthetic supervision and sufficient comparative validation (e.g., against USO) still remain.

**Reviewer Concerns:**

**Well addressed:**
1. Evaluation of Destylization Quality: The reviewers requested quantitative evidence that DestyleNet actually works. The authors added a new evaluation (Table 6) using QwenVL-Max across 5 independent trials to measure ID Score, Style Removal Score, and Image Quality. DF8q acknowledged this as a good step. (DF8q, pxZ2)
2. Dataset Imbalance and Authentic Supervision: The authors clarified that authentic supervision refers to using unmodified style images (whether real or FLUX-generated) as targets, avoiding the artifacts of pseudo-stylized targets used in prior work like OmniStyle. They also justified FLUX usage for scalability. jeZr accepted this explanation and decided to increase their rating. (jeZr)
3. Comparison with Forward-Based Pipelines: The authors clarified the fundamental difference: forward pipelines use pseudo-targets (stylized outputs) that contain artifacts, whereas this method uses clean style images as supervision. (pxZ2)
4. Generalizability across Model Scales: In response to requests for SDXL evidence, authors provided results fine-tuning SD3-Medium (2B), showing the dataset works across architectures and outperforms larger baselines. (pxZ2)

**Partly addressed:**
1. Inherited Biases from Training Data: The authors argued that the DestyleCoT-Filter removes low-quality samples and that their method statistically outperforms the baselines used for data generation. While DF8q found the rebuttal to be a good first step, NvXS remained unconvincing regarding the dependence on SOTA methods. (DF8q, NvXS)

**Unsolved:**
1. Comparison with USO: NvXS requested a direct comparison with the destylization in USO. The authors explained that USO is a concurrent work without public code/details, making direct comparison infeasible, and instead compared downstream stylization. NvXS found this unconvincing and maintained their score. (NvXS)
2. Handling Abstract Art: While the authors argued their method is compatible with abstract styles and provided visual examples, NvXS remained unconvinced about the theoretical limitation of structure-preserving destylization for abstract art. (NvXS)

**Reviewer Scores:**

**pxZ2 (4):**

This reviewer concerned on the significant difference from OmniStyle/CSGO despite the rebuttal and felt the reliance on written descriptions was not fully convincing initially. While the extensive new data addresses their core evidence concern, the concerns regarding the method's fundamental novelty remains. The score is likely to remain at 4.

**jeZr (4):**

While the authors confirmed revisions were made, the fundamental concern about the dataset's synthetic nature may limit the score increase.

**NvXS (6):**

This reviewer explicitly stated after the rebuttal: ``I find the replies to W2 and W4 unconvincing and have therefore decided to maintain my original rating.``  The score would likely remain at 6.

**DF8q (6):**

This reviewer acknowledged the rebuttal and the new quantitative experiments on destylization, stating the paper made a ``good first step``. As their main request for destylization analysis was met, the score is likely to remain at 6.

---

### Decision · Program_Chairs · 2026-01-26

Reject